# Dual-Solver: A Generalized ODE Solver for Diffusion Models with Dual Prediction

**Soochul Park**
SteAI
MODULABS
scpark20@gmail.com

**Yeon Ju Lee**
Korea University
MODULABS
leeyeonju08@korea.ac.kr

## Abstract

Diffusion models achieve state-of-the-art image quality. However, sampling is costly at inference time because it requires a large number of function evaluations (NFEs). To reduce NFEs, classical ODE numerical methods have been adopted. Yet, the choice of prediction type and integration domain leads to different sampling behaviors. To address these issues, we introduce Dual-Solver, which generalizes multistep samplers through learnable parameters that continuously (i) interpolate among prediction types, (ii) select the integration domain, and (iii) adjust the residual terms. It retains the standard predictor-corrector structure while preserving second-order local accuracy. These parameters are learned via a classification-based objective using a frozen pretrained classifier (e.g., MobileNet or CLIP). For ImageNet class-conditional generation (DiT, GM-DiT) and text-to-image generation (SANA, PixArt-$\alpha$), Dual-Solver improves FID and CLIP scores in the low-NFE regime ($3 \leq \text{NFE} \leq 9$) across backbones.

## 1 Introduction

Generative modeling aims to learn a data distribution and generate new samples that resemble real data. Classic approaches include autoregressive models that factorize likelihoods over pixels or tokens (Van den Oord et al., 2016; Salimans et al., 2017), variational auto-encoders that optimize an evidence lower bound (Kingma & Welling, 2013; Vahdat & Kautz, 2020), flow-based models that construct exact, invertible density maps (Dinh et al., 2014; Kingma & Dhariwal, 2018), and generative adversarial networks that learn via adversarial training between a generator and a discriminator (Goodfellow et al., 2020; Arjovsky et al., 2017). Diffusion models have emerged as a powerful class of generative models. In seminal work, Sohl-Dickstein et al. (2015) introduced diffusion probabilistic modeling with a forward noising process and a learned reverse process trained by minimizing a KL-divergence objective. Subsequent reformulations (Ho et al., 2020) have streamlined training and inference with simple denoising objectives (e.g., noise or data prediction), leading to state-of-the-art fidelity and robust scaling. Today, diffusion models drive progress across multiple modalities, including images (Dhariwal & Nichol, 2021; Rombach et al., 2022), audio (Kong et al., 2020; Liu et al., 2023), and video (Ho et al., 2022; Kong et al., 2024).

Diffusion models generate samples through iterative denoising steps, making the inference cost scale with the number of function evaluations (NFEs). Leveraging the probability–flow formulation, which casts sampling as an ordinary differential equation (Song et al., 2021b), a number of recent works have proposed ODE-based acceleration methods. Along one axis, classical ODE methods—singlestep Runge–Kutta (Runge, 1895) and multistep Adams–Bashforth (Bashforth & Adams, 1883)—provide off-the-shelf accuracy–NFE trade-offs for a given evaluation budget (Butcher, 2016). Along a second axis, diffusion-dedicated solvers exploit the structure of the denoising dynamics: they approximate noise or data predictions with low-order Taylor expansions or Lagrange interpolation and derive closed-form updates (Lu et al., 2022a;b; Qinsheng & Chen, 2023; Zhao et al., 2023; Xue et al., 2024). Lastly, learned solvers optimize the timestep schedule and other sampling-related parameters (Zhou et al., 2024; Shaul et al., 2023; 2024; Wang et al., 2025). Because these parameters depend on the backbone and dataset, such solvers are typically tailored to specific backbones and settings (e.g., a chosen NFE and guidance scale). Training also

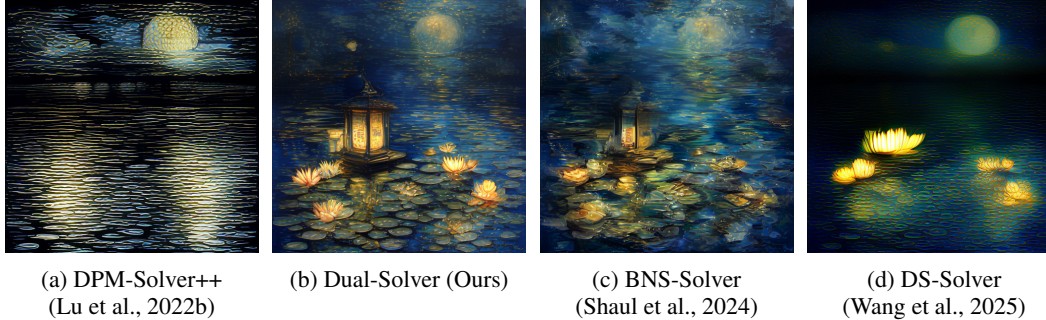

| (a) DPM-Solver++ | (b) Dual-Solver (Ours) | (c) BNS-Solver | (d) DS-Solver |
| (Lu et al., 2022b) | | (Shaul et al., 2024) | (Wang et al., 2025) |

Figure 1: **Sampling results.** SANA (Xie et al., 2024), NFE=3, CFG=4.5. See Fig. 18 for further results.

incurs substantial preparation overhead, as it requires many teacher trajectories or final samples generated using high-NFE sampling. However, compared to the classical methods and dedicated solvers discussed above, learned solvers deliver substantially better sample quality and remain an active area of research. We introduce Dual-Solver, a learned solver for diffusion models with three types of learnable parameters:

- a prediction parameter $\gamma$ that interpolates among noise, velocity, and data predictions (Sec. 3.1);
- a domain change parameter $\tau$ that interpolates between logarithmic and linear domains (Sec. 3.2);
- a residual parameter $\kappa$ that adjusts the residual term while preserving second-order accuracy (Sec. 3.3).

We further propose a classification-based learning strategy that yields high-fidelity images even in the low-NFE regime (Sec. 5.2). Unlike regression-based learning, which typically requires many target samples generated at high NFEs, our approach requires no target samples. Solver parameters are learned using either pretrained image classifiers (He et al., 2016; Dosovitskiy et al., 2020; Howard et al., 2017) or a zero-shot classifier (Radford et al., 2021). For $3 \leq \text{NFE} \leq 9$, Dual-Solver outperforms prior state-of-the-art solvers (Sec. 6).

## 2 PRELIMINARIES

### 2.1 DIFFUSION MODELS

**Training process.** Diffusion models (Ho et al., 2020; Song et al., 2021b) are trained as follows. A clean sample $\boldsymbol{x}_0$ is drawn from the data distribution, and noise is drawn independently from the standard Gaussian distribution. They are then linearly combined with signal rate $\alpha_t$ and noise rate $\sigma_t$:

$$\boldsymbol{x}_t = \alpha_t \boldsymbol{x}_0 + \sigma_t \boldsymbol{\epsilon}, \qquad \text{where } \boldsymbol{x}_0 \sim p_{\text{data}}, \ \boldsymbol{\epsilon} \sim \mathcal{N}(\boldsymbol{0}, \boldsymbol{I}), \ 0 \leq t \leq T. \tag{1}$$

Here, $\alpha_t$ and $\sigma_t$ are set by a predefined schedule. Common choices include variance-preserving (VP), with $\alpha_t^2 + \sigma_t^2 = 1$ (Sohl-Dickstein et al., 2015; Ho et al., 2020); variance-exploding (VE), with $\alpha_t = 1$, $\sigma_t \geq 0$ (Song & Ermon, 2019; 2020); and optimal transport (OT), with $\alpha_t = 1 - t$, $\sigma_t = t$, $T = 1$ (Lipman et al., 2022). The backbone is trained to take a noisy sample $\boldsymbol{x}_t$ and the time $t$ as input and to predict one of the following: the noise $\boldsymbol{\epsilon}$, the clean sample $\boldsymbol{x}_0$ (Ho et al., 2020), or the velocity $\boldsymbol{v}_t = \frac{d\alpha_t}{dt}\boldsymbol{x}_0 + \frac{d\sigma_t}{dt}\boldsymbol{\epsilon}$ (Lipman et al., 2022). By convention, the backbone parameters are denoted $\theta$, and the predictions are written as $\boldsymbol{\epsilon}_\theta$ (noise), $\boldsymbol{x}_\theta$ (data), and $\boldsymbol{v}_\theta$ (velocity).

**Sampling process.** The dynamics corresponding to Eq. 1 can be expressed as the following stochastic differential equation (SDE) (Song et al., 2021b):

$$d\boldsymbol{x}_t = f_t \boldsymbol{x}_t \, dt + g_t \, d\boldsymbol{w}_t, \qquad f_t = \frac{d \log \alpha_t}{dt}, \quad g_t^2 = \frac{d\sigma_t^2}{dt} - 2 \frac{d \log \alpha_t}{dt} \sigma_t^2, \tag{2}$$

where $\boldsymbol{w}_t$ is a standard Wiener process. A probability-flow ODE that shares the same marginal distributions as the SDE at every time $t$ has been proposed (Song et al., 2021b):

$$\frac{d\boldsymbol{x}_t}{dt} = f_t\,\boldsymbol{x}_t - \frac{1}{2}g_t^2\,\nabla_{\boldsymbol{x}}\log q_t(\boldsymbol{x}_t), \quad \text{where } \nabla_{\boldsymbol{x}}\log q_t(\boldsymbol{x}_t) = -\frac{\mathbb{E}[\boldsymbol{\epsilon}\mid\boldsymbol{x}_t]}{\sigma_t} \approx -\frac{\boldsymbol{\epsilon}_\theta(\boldsymbol{x}_t,t)}{\sigma_t}. \tag{3}$$

Given this ODE, one can perform sampling using classical numerical methods beyond Euler, such as singlestep Runge–Kutta (Runge, 1895) and multistep Adams–Bashforth schemes (Bashforth & Adams, 1883). Moreover, several works split the right-hand side into linear and nonlinear parts, and evaluate the nonlinear term using finite-difference approximations (Lu et al., 2022a;b; Zhao et al., 2023) or via Lagrange interpolation (Qinsheng & Chen, 2023; Xue et al., 2024).

## 2.2 Prediction Types

Diffusion backbones are typically trained with noise prediction (Ho et al., 2020; Karras et al., 2022; Peebles & Xie, 2023), whereas flow-matching backbones are trained with velocity prediction (Lipman et al., 2022; Chen et al., 2025; Xie et al., 2024). However, noise, velocity, and data predictions can be converted into one another as follows.

$$\boldsymbol{x}_\theta(\boldsymbol{x}_t,t) = \frac{\boldsymbol{x}_t - \sigma_t\,\boldsymbol{\epsilon}_\theta(\boldsymbol{x}_t,t)}{\alpha_t}, \quad \boldsymbol{v}_\theta(\boldsymbol{x}_t,t) = \frac{d\alpha_t}{dt}\,\boldsymbol{x}_\theta(\boldsymbol{x}_t,t) + \frac{d\sigma_t}{dt}\,\boldsymbol{\epsilon}_\theta(\boldsymbol{x}_t,t). \tag{4}$$

Using these relationships, a different prediction type can be used at sampling time from that used during training. For diffusion models, many early samplers use noise prediction (Ho et al., 2020; Song et al., 2021a; Liu et al., 2022; Qinsheng & Chen, 2023; Lu et al., 2022a). More recently, data prediction has also been widely adopted to improve sample quality (Lu et al., 2022b; Xie et al., 2024; Wang et al., 2025). In contrast, flow-matching backbones typically use velocity prediction at sampling time, as in training (Lipman et al., 2022; Chen et al., 2025; Xie et al., 2024; Shaul et al., 2024). Table 1 summarizes the sampling equations in both differential and integral form for each prediction type, derived from Eq. 3 and Eq. 4.

**Discretization discrepancy.** We ask a simple question: do the prediction types yield the same update? In continuous time, yes; in discrete time, no. Fig. 2 illustrates this discrepancy in two dimensions. For example, applying the Euler method in the $\lambda$-domain (half log-SNR) to noise prediction yields $\boldsymbol{x}_{t_{i+1}}^{\text{noise}} = \frac{\alpha_{t_{i+1}}}{\alpha_{t_i}}\big[\boldsymbol{x}_{t_i} - \sigma_{t_i}\,\Delta\lambda_{t_i}\,\boldsymbol{\epsilon}_\theta(\boldsymbol{x}_{t_i},t_i)\big]$, where $\Delta\lambda_{t_i} := \lambda_{t_{i+1}} - \lambda_{t_i}$. Applying the same method to data prediction and rewriting the result using Eq. 4 gives $\boldsymbol{x}_{t_{i+1}}^{\text{data}} = \frac{\alpha_{t_{i+1}}}{\alpha_{t_i}}\big[(1+\Delta\lambda_{t_i})e^{-\Delta\lambda_{t_i}}\,\boldsymbol{x}_{t_i} - e^{-\Delta\lambda_{t_i}}\sigma_{t_i}\,\Delta\lambda_{t_i}\,\boldsymbol{\epsilon}_\theta(\boldsymbol{x}_{t_i},t_i)\big]$. Since $e^{-\Delta\lambda_{t_i}} = 1 - \Delta\lambda_{t_i} + \frac{1}{2}\Delta\lambda_{t_i}^2 + \cdots$, they differ at order $O((\Delta\lambda_{t_i})^2)$. This naturally raises the question of which update is preferable in practice.

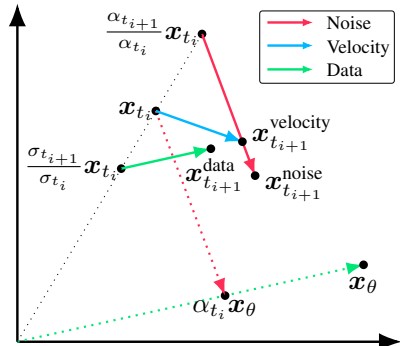

Figure 2: Euler updates for noise, velocity, and data predictions.

Table 1: Differential and integral forms for noise, velocity, and data predictions. ($\alpha_t$: signal rate, $\sigma_t$: noise rate, $\lambda_t := \log(\alpha_t/\sigma_t)$)

| | Differential form | Integral form on $[t_i, t_{i+1}]$ |
|---|---|---|
| Noise | $\frac{d\boldsymbol{x}_t}{dt} = \frac{d\log\alpha_t}{dt}\,\boldsymbol{x}_t - \sigma_t\frac{d\lambda_t}{dt}\,\boldsymbol{\epsilon}_\theta(\boldsymbol{x}_t,t)$ | $\boldsymbol{x}_{t_{i+1}} = \frac{\alpha_{t_{i+1}}}{\alpha_{t_i}}\boldsymbol{x}_{t_i} - \alpha_{t_{i+1}}\int_{t_i}^{t_{i+1}}\frac{\sigma_t}{\alpha_t}\frac{d\lambda_t}{dt}\,\boldsymbol{\epsilon}_\theta(\boldsymbol{x}_t,t)\,dt$ |
| Velocity | $\frac{d\boldsymbol{x}_t}{dt} = \frac{d\alpha_t}{dt}\,\boldsymbol{x}_\theta(\boldsymbol{x}_t,t) + \frac{d\sigma_t}{dt}\,\boldsymbol{\epsilon}_\theta(\boldsymbol{x}_t,t)$ | $\boldsymbol{x}_{t_{i+1}} = \boldsymbol{x}_{t_i} + \int_{t_i}^{t_{i+1}}\big[\frac{d\alpha_t}{dt}\,\boldsymbol{x}_\theta(\boldsymbol{x}_t,t) + \frac{d\sigma_t}{dt}\,\boldsymbol{\epsilon}_\theta(\boldsymbol{x}_t,t)\big]\,dt$ |
| Data | $\frac{d\boldsymbol{x}_t}{dt} = \frac{d\log\sigma_t}{dt}\,\boldsymbol{x}_t + \alpha_t\frac{d\lambda_t}{dt}\,\boldsymbol{x}_\theta(\boldsymbol{x}_t,t)$ | $\boldsymbol{x}_{t_{i+1}} = \frac{\sigma_{t_{i+1}}}{\sigma_{t_i}}\boldsymbol{x}_{t_i} + \sigma_{t_{i+1}}\int_{t_i}^{t_{i+1}}\frac{\alpha_t}{\sigma_t}\frac{d\lambda_t}{dt}\,\boldsymbol{x}_\theta(\boldsymbol{x}_t,t)\,dt$ |

## 3 DUAL-SOLVER

### 3.1 DUAL PREDICTION WITH PARAMETER $\gamma$

We propose a *dual prediction* scheme that uses both $\boldsymbol{x}_\theta$ and $\boldsymbol{\epsilon}_\theta$ together. We call it *dual* because it treats $\boldsymbol{x}_\theta$ and $\boldsymbol{\epsilon}_\theta$ separately, unlike velocity prediction, which bundles them into $\boldsymbol{v}_\theta$. Moreover, we introduce the following integral formulation parameterized by $\gamma$, which interpolates between the integral forms of noise, velocity, and data prediction.

**Integral form of dual prediction.**

$$\boldsymbol{x}_{t_{i+1}} = A\,\boldsymbol{x}_{t_i} + B\left[\int_{t_i}^{t_{i+1}} C\,\boldsymbol{x}_\theta(\boldsymbol{x}_t, t)\,dt + \int_{t_i}^{t_{i+1}} D\,\boldsymbol{\epsilon}_\theta(\boldsymbol{x}_t, t)\,dt\right],$$

$$\text{where}\quad \begin{cases} A = \left(\sigma_{t_{i+1}}/\sigma_{t_i}\right)^\gamma,\ B = \sigma_{t_{i+1}}^\gamma,\ C = \dfrac{d}{dt}(\alpha_t\,\sigma_t^{-\gamma}),\ D = \dfrac{d}{dt}(\sigma_t^{1-\gamma}) \text{ for } \gamma \geq 0, \\[2mm] A = \left(\alpha_{t_{i+1}}/\alpha_{t_i}\right)^{-\gamma},\ B = \alpha_{t_{i+1}}^{-\gamma},\ C = \dfrac{d}{dt}(\alpha_t^{1+\gamma}),\ D = \dfrac{d}{dt}(\sigma_t\,\alpha_t^\gamma) \text{ for } \gamma < 0. \end{cases} \quad (5)$$

It has the following properties:

- When $\gamma = -1$, the integral reduces to the noise-prediction form (Table 1).
- When $\gamma = 0$, the integral reduces to the velocity-prediction form (Table 1).
- When $\gamma = 1$, the integral reduces to the data-prediction form (Table 1).

From Eq. 3, we derive the differential form of dual prediction and, by integrating, obtain the integral form given above; the full derivation is provided in Appendix B.1. Another role of $\gamma$ is to control how much $\boldsymbol{x}_{t_i}$ contributes to $\boldsymbol{x}_{t_{i+1}}$: increasing $\gamma$ decreases the coefficient $A$ multiplying $\boldsymbol{x}_{t_i}$, whereas decreasing $\gamma$ increases $A$.

### 3.2 LOG-LINEAR DOMAIN CHANGE WITH PARAMETER $\tau$

**Domain change.** Let $L : (0, \infty) \to \mathcal{I}$ be a $C^1$ diffeomorphism onto an interval $\mathcal{I} \subseteq \mathbb{R}$. To apply a change of variables to the integrals in Eq. 5, we define the following:

$$\begin{cases} u(t) = L\left(\alpha_t\,\sigma_t^{-\gamma}\right),\ v(t) = L\left(\sigma_t^{1-\gamma}\right) \text{ for } \gamma \geq 0, \\[1mm] u(t) = L\left(\alpha_t^{1+\gamma}\right),\ v(t) = L\left(\sigma_t\,\alpha_t^\gamma\right) \text{ for } \gamma < 0. \end{cases} \quad (6)$$

By the chain rule, $\frac{d}{dt}L^{-1}(u) = \frac{dL^{-1}(u)}{du}\frac{du}{dt}$ and similarly for $v$. Thus, we obtain

$$\boldsymbol{x}_{t_{i+1}} = A\boldsymbol{x}_{t_i} + B\left[\int_{u_i}^{u_{i+1}} \frac{dL^{-1}(u)}{du}\,\boldsymbol{x}_\theta(u)\,du + \int_{v_i}^{v_{i+1}} \frac{dL^{-1}(v)}{dv}\,\boldsymbol{\epsilon}_\theta(v)\,dv\right]. \quad (7)$$

Here, $u_i = u(t_i)$, $u_{i+1} = u(t_{i+1})$ and $v_i = v(t_i)$, $v_{i+1} = v(t_{i+1})$. $A$ and $B$ are as defined in Eq. 5, and $\boldsymbol{x}_\theta(u) \coloneqq \boldsymbol{x}_\theta(\boldsymbol{x}_{u^{-1}(u)}, u^{-1}(u))$ and $\boldsymbol{\epsilon}_\theta(v) \coloneqq \boldsymbol{\epsilon}_\theta(\boldsymbol{x}_{v^{-1}(v)}, v^{-1}(v))$.

**Log-linear transform.** Previous works (Dockhorn et al., 2022; Qinsheng & Chen, 2023; Zhou et al., 2024) use the linear transform $L(y) = y$ with noise prediction. Because $\frac{d}{du}L^{-1}(u) = 1$, the integrand carries no weighting factor, making it straightforward to develop approximations such as Taylor expansions and Lagrange interpolation. By contrast, other works (Lu et al., 2022a;b; Zhao et al., 2023; Xue et al., 2024) use a logarithmic transform $L(y) = \log y$. Because $\frac{d}{du}L^{-1}(u) = e^u$, the integrand carries an exponential weight. A closed-form approximation can be obtained via an exponential integrator (Hochbruck & Ostermann, 2010) or by using Lagrange interpolation. Motivated by these works, we propose a *log-linear* transform parameterized by a scalar $\tau$.

$$L(y; \tau) = \frac{\log(1 + \tau y)}{\tau}, \quad \tau > 0. \quad (8)$$

This transform is invertible, with inverse $L^{-1}(u; \tau) = \left(e^{\tau u} - 1\right)/\tau$; its weighting factor is $\frac{d}{du}L^{-1}(u; \tau) = e^{\tau u}$. Consequently, it has the following properties:

- As $\tau \to 0^+$ : $L(y; \tau) \to y$, $\frac{d}{du}L^{-1}(u; \tau) \to 1$.
- When $\tau = 1$ : $L(y; \tau) = \log(1 + y)$, $\frac{d}{du}L^{-1}(u; \tau) = e^u$.

We apply the log–linear transform to Eq. 7, allowing separate parameters $\tau_u$ and $\tau_v$ for the $u$- and $v$-integrals, respectively. A comparison of domain-change functions is presented in Sec. 6.2.3.

### 3.3 Approximation with Parameter $\kappa$

On the interval $[u_i, u_{i+1}]$, we approximate $\boldsymbol{x}_\theta$ in two ways: (i) the first-order approximation $\boldsymbol{x}_\theta(u) = \boldsymbol{x}_\theta(u_i)$; and (ii) the second-order forward-difference approximation $\boldsymbol{x}_\theta(u) = \boldsymbol{x}_\theta(u_i) + \frac{\Delta \boldsymbol{x}_\theta(u_i)}{\Delta u_i}(u - u_i)$. Here, $\boldsymbol{x}_\theta(u_i) := \boldsymbol{x}_\theta(\boldsymbol{x}_{u^{-1}(u_i)}, u^{-1}(u_i))$, $\Delta \boldsymbol{x}_\theta(u_i) := \boldsymbol{x}_\theta(u_{i+1}) - \boldsymbol{x}_\theta(u_i)$, and $\Delta u_i := u_{i+1} - u_i$. We also introduce $K(\Delta u_i; \kappa_u) = \kappa_u (\Delta u_i)^2$, an $\mathcal{O}((\Delta u_i)^2)$ term, to allow additional flexibility in the residual term while preserving local accuracy. $\kappa_u$ is a real scalar parameter that controls the magnitude of the residual term. Applying the same approximations to $\boldsymbol{\epsilon}_\theta$ yields the following first-order predictor $\boldsymbol{x}_{t_{i+1}}^{\text{1st-pred.}}$ and second-order corrector $\boldsymbol{x}_{t_{i+1}}^{\text{2nd-corr.}}$:

$$\boldsymbol{x}_{t_{i+1}}^{\text{1st-pred.}} = A\boldsymbol{x}_{t_i} + B\Big[\boldsymbol{x}_\theta(u_i)\big(\Delta L^{-1}(u_i) + K(\Delta u_i; \kappa_u)\big) + \boldsymbol{\epsilon}_\theta(v_i)\big(\Delta L^{-1}(v_i) + K(\Delta v_i; \kappa_v)\big)\Big]. \tag{9}$$

$$\boldsymbol{x}_{t_{i+1}}^{\text{2nd-corr.}} = A\boldsymbol{x}_{t_i} + B\Big[\boldsymbol{x}_\theta(u_i)\,\Delta L^{-1}(u_i) + \frac{\Delta \boldsymbol{x}_\theta(u_i)}{2}\big(\Delta L^{-1}(u_i) + K(\Delta u_i; \kappa_u)\big) + \boldsymbol{\epsilon}_\theta(v_i)\,\Delta L^{-1}(v_i) + \frac{\Delta \boldsymbol{\epsilon}_\theta(v_i)}{2}\big(\Delta L^{-1}(v_i) + K(\Delta v_i; \kappa_v)\big)\Big]. \tag{10}$$

Here, $\Delta L^{-1}(u_i) := L^{-1}(u_{i+1}) - L^{-1}(u_i)$ and $A$ and $B$ are as defined in Eq. 5. When evaluating $L^{-1}(u_i)$, with $\alpha_{t_i}$, $\sigma_{t_i}$, and $\gamma$ already known, it can be obtained without writing $L^{-1}$ explicitly. For example, for $\gamma \geq 0$, we have $L^{-1}(u_i) = \alpha_{t_i} \sigma_{t_i}^{-\gamma}$ (see Eq. 6).

Detailed derivations of the predictor and corrector are provided in Appendix B.2, and the corresponding local-accuracy theorem and its proof are given in Appendix C.

## 4 Implementation Details

Dual-Solver performs sampling using a predictor–corrector scheme (Butcher, 2016) based on the equations developed in the previous section. We detail the sampling scheme in Sec. 4.1 and then present all learnable parameters in Sec. 4.2.

### 4.1 Sampling Scheme

Alg. 1 describes the sampling procedure of Dual-Solver. Sampling requires a backbone that provides both $\boldsymbol{x}_\theta$ and $\boldsymbol{\epsilon}_\theta$; when only one prediction is available, the other can be obtained via Eq. 4. Given $M$ steps, we use timesteps $\{t_i\}_{i=0}^M$ with $t_0 = T$ and draw the initial noise $\boldsymbol{x}_{t_0} \sim \mathcal{N}(0, I)$. We also use a list $\ell$ to store previous evaluations. Empirically, a first-order predictor with a second-order corrector performs best, as shown in Sec. 6.2. At step $i$, the first-order predictor takes the current state $\boldsymbol{x}_{t_i}$ and the model evaluations $\{\boldsymbol{x}_\theta(\boldsymbol{x}_{t_i}, t_i), \boldsymbol{\epsilon}_\theta(\boldsymbol{x}_{t_i}, t_i)\}$ to produce a provisional sample $\boldsymbol{x}'_{t_{i+1}}$. The second-order corrector then combines the evaluations at $t_i$ with fresh evaluations at $t_{i+1}$, i.e., $\{\boldsymbol{x}_\theta(\boldsymbol{x}'_{t_{i+1}}, t_{i+1}), \boldsymbol{\epsilon}_\theta(\boldsymbol{x}'_{t_{i+1}}, t_{i+1})\}$, to yield the corrected sample $\boldsymbol{x}_{t_{i+1}}$. At the final step $i = M - 1$, the corrector is not applied.

### 4.2 Learnable Parameters

For each $i$-th step, the parameter sets are $\phi_i^{\text{pred}} = \{\gamma_i^{\text{pred}}, \tau_{u,i}^{\text{pred}}, \tau_{v,i}^{\text{pred}}, \kappa_{u,i}^{\text{pred}}, \kappa_{v,i}^{\text{pred}}\}$ and $\phi_i^{\text{corr}} = \{\gamma_i^{\text{corr}}, \tau_{u,i}^{\text{corr}}, \tau_{v,i}^{\text{corr}}, \kappa_{u,i}^{\text{corr}}, \kappa_{v,i}^{\text{corr}}\}$. Thus, each step uses $2 \times 5 = 10$ parameters, except for the last step ($i = M - 1$), which does not use the corrector and therefore has only 5 parameters. Fig. 3 shows the learned parameters for the NFE=5 setting using a DiT (Peebles & Xie, 2023) backbone. Assuming a noise-prediction backbone (the same reasoning applies to data- and velocity-prediction backbones) and a first-order predictor with a second-order corrector (Sec. 4.1), $\phi_i^{\text{pred}}$ and $\phi_i^{\text{corr}}$ determine the coefficients for an update that combines the current state $\boldsymbol{x}_{t_i}$ and two model evaluations, $\boldsymbol{\epsilon}_\theta(\boldsymbol{x}'_{t_i}, t_i)$ and $\boldsymbol{\epsilon}_\theta(\boldsymbol{x}'_{t_{i+1}}, t_{i+1})$, to produce $\boldsymbol{x}_{t_{i+1}}$. This may seem heavy, but Sec. 6.2.2 shows it is necessary.

In addition to these parameters, we also learn the evaluation times $\{t_i\}_{i=1}^{M-1}$ (with $t_0$ and $t_M$ fixed), where $M$ denotes the number of steps. Following prior work (Tong et al., 2025; Wang et al., 2025), we employ unnormalized step variables $\{\Delta t'_i\}_{i=0}^{M-1}$ and apply a softmax over $i = 0, \ldots, M - 1$ to obtain normalized step sizes $\{\Delta t_i\}_{i=0}^{M-1}$ (nonnegative and summing to one). The timesteps are obtained via a cumulative sum: $t_i = t_0 + (t_M - t_0) \sum_{k=0}^{i-1} \Delta t_k$, $i = 1, \ldots, M - 1$.

**Algorithm 1** Dual-Solver predictor–corrector sampling (Sec. 4.1)

---

**Require:** Diffusion backbone with dual prediction $\{\boldsymbol{x}_\theta, \boldsymbol{\epsilon}_\theta\}$, timesteps $\{t_i\}_{i=0}^{M}$, initial noise $\boldsymbol{x}_{t_0}$, empty list $\ell$, parameters $\phi$
1: Evaluate $\{\boldsymbol{x}_\theta(\boldsymbol{x}_{t_0}, t_0), \boldsymbol{\epsilon}_\theta(\boldsymbol{x}_{t_0}, t_0)\}$ and add to $\ell$
2: **for** $i = 0$ **to** $M - 1$ **do**
3:      $\boldsymbol{x}'_{t_{i+1}} \leftarrow \mathrm{Predictor}(\boldsymbol{x}_{t_i}, \ell; \phi_i^{\mathrm{pred}})$
4:      **if** $i == M - 1$ **then break**
5:      Evaluate $\{\boldsymbol{x}_\theta(\boldsymbol{x}'_{t_{i+1}}, t_{i+1}),$
        $\boldsymbol{\epsilon}_\theta(\boldsymbol{x}'_{t_{i+1}}, t_{i+1})\}$ and add to $\ell$
6:      $\boldsymbol{x}_{t_{i+1}} \leftarrow \mathrm{Corrector}(\boldsymbol{x}_{t_i}, \ell; \phi_i^{\mathrm{corr}})$
7: **end for**
8: **return** $\boldsymbol{x}'_{t_M}$

**Algorithm 2** Hard-label classification for parameter learning (Sec. 5.2)

---

**Require:** Diffusion backbone with parameters $\theta$, VAE decoder $\mathcal{D}$, pretrained classifier $\mathcal{C}$, solver $\mathcal{S}$ with parameters $\phi$, label dataset $\mathcal{Y}$, learning rate $\eta$
1: **while** not converged **do**
2:      Sample $\boldsymbol{x}_T \sim \mathcal{N}(0, I)$     ▷ initial noise
3:      Sample $y \sim \mathcal{Y}$         ▷ class label
4:      $\boldsymbol{x}_0 \leftarrow \mathcal{S}(\boldsymbol{x}_T ; y, \phi, \theta)$     ▷ sampling
5:      $\hat{\boldsymbol{x}}_0 \leftarrow \mathcal{D}(\boldsymbol{x}_0)$        ▷ decoding
6:      $p \leftarrow \mathcal{C}(\hat{\boldsymbol{x}}_0)$     ▷ class probabilities
7:      $\mathcal{L} \leftarrow \mathrm{CrossEntropy}(p, y)$     ▷ loss
8:      $\phi \leftarrow \phi - \eta \nabla_\phi \mathcal{L}$    ▷ parameter update
9: **end while**

## 5 SOLVER PARAMETER LEARNING

In this section, we review existing regression-based parameter learning methods, identify their limitations, and introduce a classification-based approach. Fig. 4 provides a schematic overview of all these methods. We apply them to the proposed Dual-Solver and report FID results for each method in Sec. 6.2.5.

### 5.1 REGRESSION-BASED PARAMETER LEARNING

In regression-based learning, a solver with trainable parameters is referred to as a student solver, while an existing fixed solver is referred to as a teacher solver. The student is then trained to imitate the behavior of the teacher running at a high NFE. Most prior works adopt *trajectory regression* (Shaul et al., 2023; Zhou et al., 2024; Wang et al., 2025), which compares the trajectories generated by the teacher and the student, or *sample regression* (Shaul et al., 2024), which compares only the final samples. Since comparisons in the trajectory or sample space often show a mismatch with visual perceptual quality, *feature regression* has been proposed (Tong et al., 2025), in which the discrepancy is measured in a feature space using metrics such as LPIPS (Zhang et al., 2018). However, all of these methods require a teacher solver and incur substantial overhead in preparing supervision targets, and they tend to perform poorly in the very low NFE regime (e.g., NFE $\leq 5$; see Table 5).

### 5.2 CLASSIFICATION-BASED PARAMETER LEARNING

Beyond feature regression, we consider *soft-label classification*, where we apply a cross-entropy loss between the classifier outputs (probabilities) of the student and the teacher. This approach yields improved results in the low-NFE regime (Table 5), but it still requires a teacher solver to generate the target probabilities.

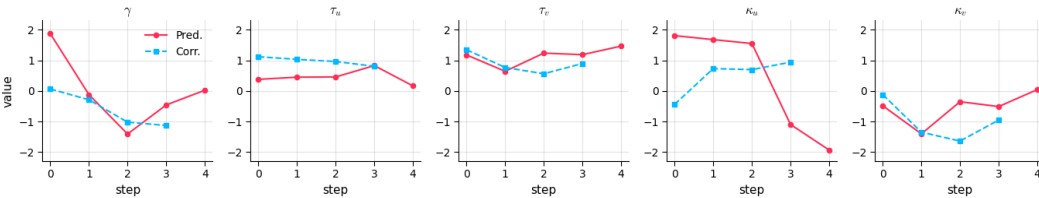

Figure 3: **Learned parameters.** Values of $\{\gamma, \tau_u, \tau_v, \kappa_u, \kappa_v\}$ across sampling steps, learned on DiT (Peebles & Xie, 2023) at NFE = 5. See Figs. 11, 12, 13, and 14 for further results.

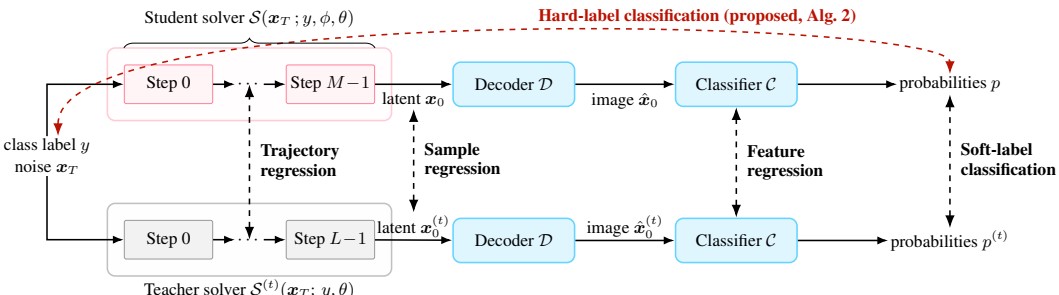

Figure 4: **Solver parameter learning methods.** It schematically illustrates trajectory, sample, and feature regression, as well as soft- and hard-label classification methods.

To remove this dependency, we further propose *hard-label classification*, which compares the student's classifier outputs (probabilities) against the input class label using a cross-entropy loss, without any teacher solver. The overall training procedure is described in Alg. 2. The sample $x_0$ generated by the solver $\mathcal{S}$ is passed through the decoder $\mathcal{D}$ and the classifier $\mathcal{C}$ to obtain class probabilities $p$. We then compute a cross-entropy loss between $p$ and the class label $y \sim \mathcal{Y}$, and update the solver parameters $\phi$ for all time steps via backpropagation. For text-to-image tasks, the class labels are replaced with text prompts, which can be obtained from datasets such as MSCOCO 2014 (Lin et al., 2014) or MJHQ-30K (Li et al., 2024), and the cross-entropy loss is replaced with CLIP loss (Radford et al., 2021).

Unlike regression to teacher targets, this method focuses on whether the generated samples lie on the correct side of the classifier's decision boundary, enabling less restrictive training. A potential issue is a mismatch between the distribution learned by the classifier and the ground-truth data distribution, but this can be mitigated by selecting an appropriate classifier. In Sec. 6.2.4, we examine the relationship between the classifier's accuracy and the resulting FID scores.

## 6 EXPERIMENTS

We benchmark Dual-Solver against two families of baselines:

- Dedicated solvers: DDIM (Song et al., 2021a), DPM-Solver++ (Lu et al., 2022b).
- Learned solvers: BNS-Solver (Shaul et al., 2024), DS-Solver (Wang et al., 2025).

We select backbones that span diffusion and flow matching, covering both ImageNet (Deng et al., 2009) conditional generation and text-to-image tasks:

- DiT-XL/2-256×256 (Peebles & Xie, 2023): diffusion, ImageNet.
- PixArt-$\alpha$ XL-2-512 (Chen et al., 2023): diffusion, text-to-image.
- GM-DiT 256×256 (Chen et al., 2025): flow matching, ImageNet.
- SANA 600M-512px (Xie et al., 2024): flow matching, text-to-image.

Solver implementations are taken from official sources or reimplemented by us. The backbones can be run via the diffusers library (von Platen et al., 2022). Further details are provided in Appendix D.

### 6.1 MAIN QUANTITATIVE RESULTS

We evaluate quantitative performance using FID (Heusel et al., 2017) and CLIP score (Radford et al., 2021). For DiT and GM-DiT, FID is computed on 50k images uniformly sampled across the 1,000 ImageNet (Deng et al., 2009) classes. For SANA and PixArt-$\alpha$, FID and CLIP are computed on the MSCOCO 2014 (Lin et al., 2014) validation set (30k image–caption pairs). The CLIP score is computed as the cosine similarity between text and image features. As described in Sec. 5, we train Dual-Solver with a classification-based objective. For DiT and GM-DiT, MobileNetV3-Large (Howard et al., 2019) is used, and for SANA and PixArt-$\alpha$, CLIP(RN101) is used. Fig. 5 shows the measured FID and CLIP scores. Across all evaluated NFEs, Dual-Solver outperforms competing solvers on both FID and CLIP for DiT, SANA, and PixArt-$\alpha$. For GM-DiT, Dual-Solver underperforms at NFE 7–9; however, when trained with a trajectory regression-based objective, it surpasses the baselines at NFE = 8 and 9 (Table 7).

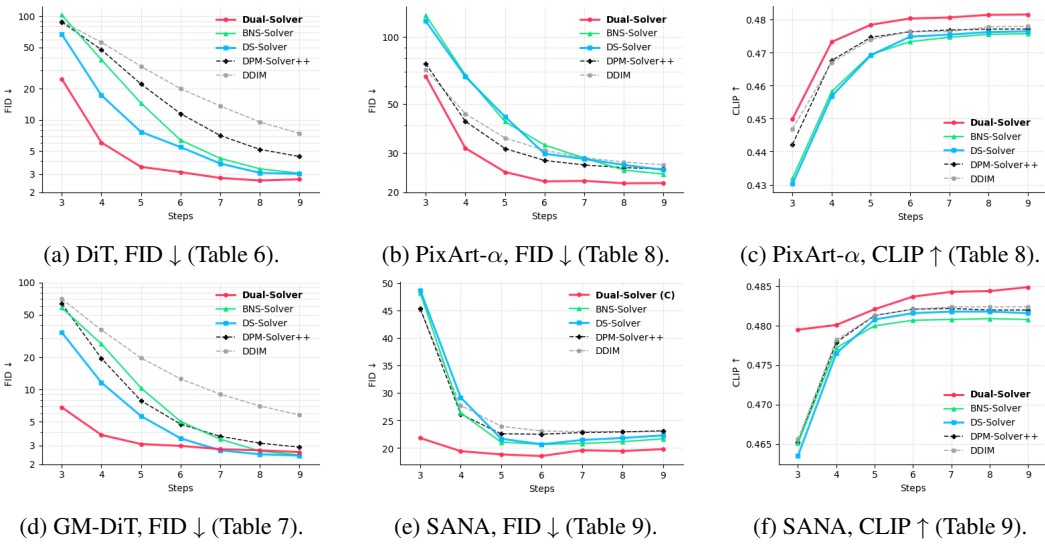

(a) DiT, FID ↓ (Table 6).     (b) PixArt-$\alpha$, FID ↓ (Table 8).     (c) PixArt-$\alpha$, CLIP ↑ (Table 8).

(d) GM-DiT, FID ↓ (Table 7).     (e) SANA, FID ↓ (Table 9).     (f) SANA, CLIP ↑ (Table 9).

Figure 5: **Main quantitative results.** FID and CLIP score; evaluated on 50k (DiT/GM-DiT) and 30k (SANA/PixArt-$\alpha$) samples; CFG: DiT=1.5, GM-DiT=1.4, SANA=4.5, PixArt-$\alpha$=3.5.

## 6.2 ABLATION STUDY

### 6.2.1 PREDICTOR-CORRECTOR CONFIGURATIONS

Table 2: Ablation of predictor–corrector configurations on DiT.

|  | Cross-entropy loss (↓) | | | |
|---|---|---|---|---|
| NFE | 3 | 5 | 7 | 9 |
| p1 | 0.667 | 0.225 | 0.183 | 0.175 |
| p1c2 | **0.574** | **0.197** | **0.178** | **0.173** |
| p2 | 1.023 | 0.253 | 0.222 | 0.181 |
| p2c2 | 5.009 | 0.317 | 0.203 | 0.191 |

We ablate the predictor–corrector configurations: `p1` (first-order predictor only), `p1c2` (first-order predictor + second-order corrector), `p2` (second-order predictor only), and `p2c2` (second-order predictor + second-order corrector). The equations for the first-order predictor and second-order corrector are given by Eqs. 9 and 10, respectively, and the equation for the second-order predictor is provided in Eq. 12. As shown in Table 2, `p1c2` yields the best performance across NFE = 3, 5, 7, and 9.

### 6.2.2 PARAMETER SETTINGS FOR $(\gamma, \tau, \kappa)$

We ablate the parameterization by fixing selected values or sharing parameters to reduce the degrees of freedom. Setting $\gamma$ to 1, 0, or $-1$ recovers data, velocity, and noise prediction, respectively; setting $\tau = 1$ yields the log1p transform; and setting $\kappa = 0$ removes the second-order residual term. We also tie $\tau_u = \tau_v$ and $\kappa_u = \kappa_v$ to share parameters across the integrations for $\boldsymbol{x}_\theta$ and $\boldsymbol{\epsilon}_\theta$. As shown in Table 3, leaving all parameters free yields the best performance at NFE = 3 and 5. Configurations that fix or share certain parameters occasionally perform slightly better at NFE = 7 and 9.

Table 3: Ablation of the $(\gamma, \tau, \kappa)$ parameter settings on DiT.

|  | Cross-entropy loss (↓) | | | |
|---|---|---|---|---|
| NFE | 3 | 5 | 7 | 9 |
| $\gamma = 1$ fixed | 0.816 | 0.223 | 0.182 | 0.176 |
| $\gamma = 0$ fixed | 0.600 | 0.202 | 0.183 | 0.180 |
| $\gamma = -1$ fixed | 7.871 | 7.676 | 0.238 | 0.196 |
| $\tau = 1$ fixed | 0.601 | 0.217 | **0.175** | 0.178 |
| $\kappa = 0$ fixed | 0.944 | 0.256 | 0.202 | 0.190 |
| $\tau, \kappa$ shared | 0.667 | 0.221 | 0.177 | **0.169** |
| all learnable | **0.574** | **0.197** | 0.178 | 0.173 |

### 6.2.3 DOMAIN-CHANGE FUNCTION CHOICE

Table 4: Ablation of Domain Change Functions on DiT.

|  | FID (↓) | | | |
|---|---|---|---|---|
| NFE | 3 | 5 | 7 | 9 |
| Type 1 | **23.38** | 4.09 | 3.56 | 3.64 |
| Type 2 | 24.91 | **3.52** | **2.75** | **2.67** |

As candidates for the domain-change function, we consider Type 1, which linearly interpolates between the linear and logarithmic transforms: $L(y; \tau) = (1 - \tau)y + \tau \log y$, and Type 2, defined in Eq. 8. In Table 4, Type 2 achieves noticeably better results at NFE = 5, 7, and 9. According to Eq. 6, the argument $y$ of the function $L$ can become very close to zero; in this regime, the log function diverges, whereas log1p is numerically stable near zero, and we attribute the improved performance to this stability.

### 6.2.4 CLASSIFICATION MODEL SELECTION

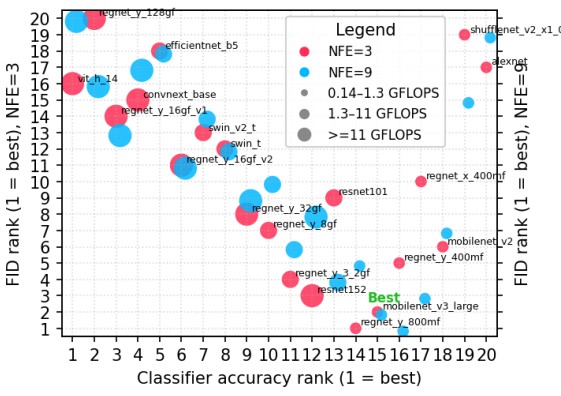

Figure 6: Classifier Accuracy vs. FID on GM-DiT.

A natural question for classification-based learning is which classifier to use. To answer this, we evaluate 20 pretrained classifiers from TorchVision (maintainers & contributors, 2016). Fig. 6 shows FID (on 10k samples) versus accuracy for each model, ordered by rank. The curve exhibits a clear V-shape: as accuracy decreases, FID improves, but beyond ranks 14–16 the FID degrades sharply. This suggests that neither very high nor very low classifier accuracy is optimal; a moderate level is most beneficial for FID. A detailed analysis linking this pattern to precision and recall (Kynkäänniemi et al., 2019) is provided in Appendix E.

### 6.2.5 PARAMETER LEARNING METHODS

Table 5 reports the results of applying the regression- and classification-based parameter learning methods discussed in Sec. 5 to Dual-Solver. Feature regression is implemented using LPIPS (Zhang et al., 2018), and it generally outperforms trajectory- and sample-space regression. The improvement is particularly pronounced at NFE = 3 and 5. Depending on which classifier (AlexNet (Krizhevsky et al., 2012), VGG (Simonyan & Zisserman, 2014), or SqueezeNet (Iandola et al., 2016)) is used to extract features, the results vary significantly, underscoring the importance of classifier choice in feature space. Classification-based methods further improve performance over feature regression at NFE = 3 and 5. In particular, the method that uses hard labels achieves the best results across all NFEs. We attribute this to the choice of an appropriate classifier, as discussed in Sec. 6.2.4.

Table 5: Comparison of parameter learning methods on DiT.

| | FID (↓) | | | |
|---|---|---|---|---|
| **NFE** | **3** | **5** | **7** | **9** |
| **Regression-based learning** (Sec. 5.1) | | | | |
| Sample | 107.13 | 11.71 | 4.60 | 2.99 |
| Trajectory | 100.89 | 11.59 | 3.66 | 2.84 |
| Feature (AlexNet) | 47.75 | 7.24 | 3.42 | 2.91 |
| Feature (VGG) | 41.58 | 5.48 | 3.23 | 2.88 |
| Feature (SqueezeNet) | 44.00 | 7.07 | 3.31 | 2.79 |
| **Classification-based learning** (Sec. 5.2) | | | | |
| Soft-label | 25.13 | 4.90 | 3.37 | 3.01 |
| Hard-label | **24.91** | **3.52** | **2.75** | **2.67** |

### 6.3 PARAMETER INTERPOLATION ACROSS NFES

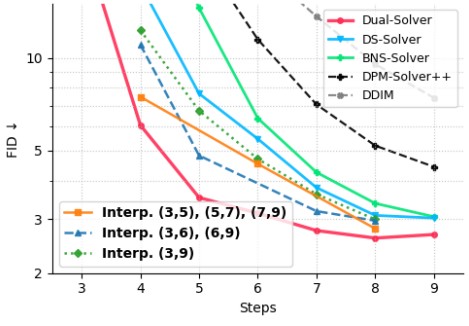

Figure 7: FID results for parameter interpolation across NFEs with DiT.

From the learned parameters of Dual-Solver (Appendix F), we observe that their overall shapes remain similar across different NFEs. Motivated by this, we test the robustness of the learned parameters by applying them to different NFEs. To obtain the parameters for an unseen NFE, we interpolate between the parameters of its two neighboring NFEs. The detailed formulas are provided in Appendix G. Fig. 7 shows the results. *Interp. (3,5), (5,7), (7,9)* denotes that we interpolate between the parameters learned at NFEs (3,5), (5,7), and (7,9) to obtain the parameters for the intermediate NFEs 4, 6, and 8, respectively. The other interpolation schemes are defined analogously. Although these interpolated parameters do not match the performance of parameters directly optimized for each NFE, the gaps are modest, and the resulting FID scores still outperform those of other solvers.

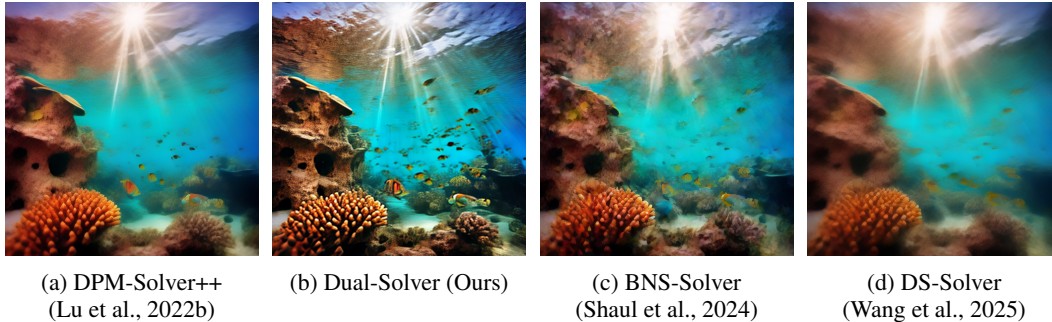

| (a) DPM-Solver++ | (b) Dual-Solver (Ours) | (c) BNS-Solver | (d) DS-Solver |
| (Lu et al., 2022b) | | (Shaul et al., 2024) | (Wang et al., 2025) |

Figure 8: **Sampling results.** PixArt-$\alpha$ (Chen et al., 2023), NFE=5, CFG=3.5. See Fig. 19 for further results.

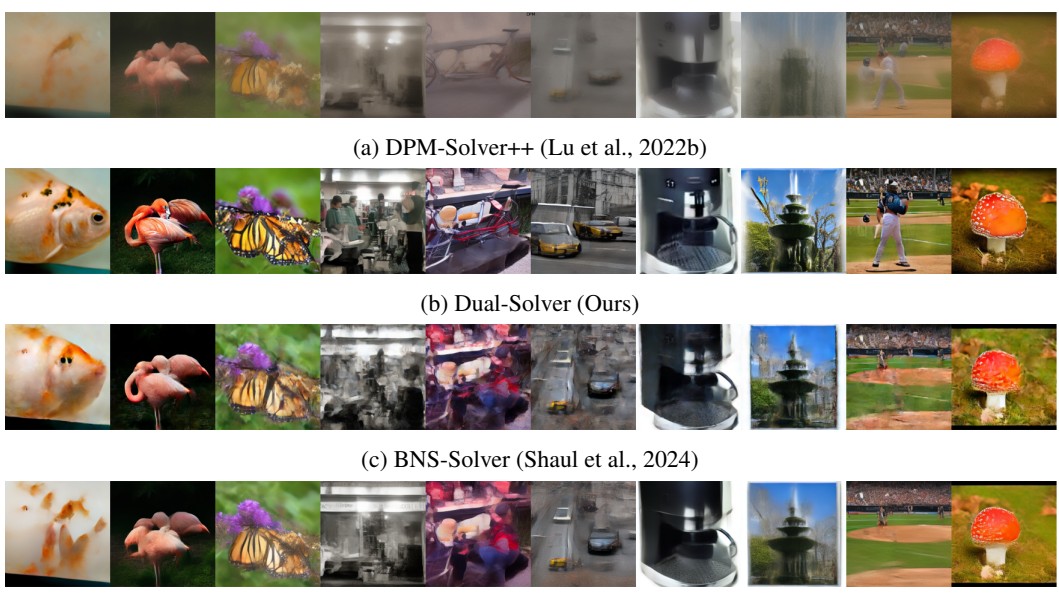

(a) DPM-Solver++ (Lu et al., 2022b)

(b) Dual-Solver (Ours)

(c) BNS-Solver (Shaul et al., 2024)

(d) DS-Solver (Wang et al., 2025)

Figure 9: **Sampling results.** GM-DiT (Chen et al., 2025), NFE=3, CFG=1.4. See Fig. 17 for further results.

## 7 CONCLUSIONS AND LIMITATIONS

This paper introduces Dual-Solver, a predictor–corrector sampler that achieves second-order numerical accuracy. It features per-step learnable parameters $(\gamma, \tau, \kappa)$ that govern the prediction types, the integration domain, and the second-order residual adjustment. All solver parameters are optimized end-to-end with a classification-based objective using a pretrained image classifier. Across diverse diffusion and flow matching backbones, experiments show substantial improvements over competing solvers in the low-NFE regime ($3 \leq \text{NFE} \leq 9$), as measured by FID and CLIP score. Limitations include the absence of unconditional backbones and a lack of analysis beyond second-order accuracy; both are left for future work.

### ACKNOWLEDGEMENTS

We would like to thank Juhee Lee, Jiyub Shin, and Kyungdo Min for their helpful discussions and careful review of the manuscript. This research was partially supported by MODULABS. This work was also supported by the Basic Science Research Program through the National Research Foundation of Korea (NRF), funded by the Ministry of Education (Grant No. RS-2025-25424642).

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

CONTENTS

## A  LLM USAGE

We disclose that Large Language Models (LLMs) were used as assistive tools for:

- verifying the consistency of mathematical derivations,
- assisting literature search and surfacing related work,
- helping with formal writing (style, grammar, typos),
- drafting figure and table scripts.

All scientific ideas, methods, and results originate from the authors, who take full responsibility for the content of this paper.

## B  DERIVATIONS

### B.1  DERIVATION OF DUAL PREDICTION

**Differential Form of Dual Prediction.**   The differential form corresponding to the integral form in Eq. 5 is given by

$$\frac{d\boldsymbol{x}_t}{dt} = \beta\boldsymbol{x}_t + \alpha_t\Big(\frac{d\log\alpha_t}{dt} - \beta\Big)\boldsymbol{x}_\theta(\boldsymbol{x}_t, t) + \sigma_t\Big(\frac{d\log\sigma_t}{dt} - \beta\Big)\boldsymbol{\epsilon}_\theta(\boldsymbol{x}_t, t). \tag{11}$$

To derive this form, we start from the probability-flow ODE in Eq. 3:

$$\frac{d\boldsymbol{x}_t}{dt} = f_t\,\boldsymbol{x}_t - \frac{1}{2}\,g_t^2\,\nabla_{\boldsymbol{x}}\log q(\boldsymbol{x}_t),$$

$$\text{where } f_t = \frac{d\log\alpha_t}{dt},\ g_t^2 = \frac{d}{dt}\sigma_t^2 - 2\,f_t\,\sigma_t^2,\ \nabla_{\boldsymbol{x}}\log q(\boldsymbol{x}_t) = -\frac{\boldsymbol{\epsilon}_\theta(\boldsymbol{x}_t, t)}{\sigma_t}.$$

Substituting $f_t$, $g_t^2$, and $\nabla_{\boldsymbol{x}}\log q(\boldsymbol{x}_t)$ yields:

$$\begin{aligned}
\frac{d\boldsymbol{x}_t}{dt} &= \frac{d\log\alpha_t}{dt}\,\boldsymbol{x}_t + \frac{1}{2\sigma_t}\left(\frac{d}{dt}\sigma_t^2 - 2\frac{d\log\alpha_t}{dt}\,\sigma_t^2\right)\boldsymbol{\epsilon}_\theta(\boldsymbol{x}_t, t) \\
&= \frac{d\log\alpha_t}{dt}\,\boldsymbol{x}_t + \sigma_t\left(\frac{d\log\sigma_t}{dt} - \frac{d\log\alpha_t}{dt}\right)\boldsymbol{\epsilon}_\theta(\boldsymbol{x}_t, t).
\end{aligned}$$

Introducing an arbitrary $\beta \in \mathbb{R}$, we can rewrite:

$$\begin{aligned}
\frac{d\boldsymbol{x}_t}{dt} &= \beta\boldsymbol{x}_t + \left(\frac{d\log\alpha_t}{dt} - \beta\right)\boldsymbol{x}_t + \sigma_t\left(\frac{d\log\sigma_t}{dt} - \frac{d\log\alpha_t}{dt}\right)\boldsymbol{\epsilon}_\theta(\boldsymbol{x}_t, t) \\
&= \beta\boldsymbol{x}_t + \left(\frac{d\log\alpha_t}{dt} - \beta\right)\big(\alpha_t\boldsymbol{x}_\theta(\boldsymbol{x}_t, t) + \sigma_t\boldsymbol{\epsilon}_\theta(\boldsymbol{x}_t, t)\big) + \sigma_t\left(\frac{d\log\sigma_t}{dt} - \frac{d\log\alpha_t}{dt}\right)\boldsymbol{\epsilon}_\theta(\boldsymbol{x}_t, t) \\
&= \beta\boldsymbol{x}_t + \alpha_t\Big(\frac{d\log\alpha_t}{dt} - \beta\Big)\boldsymbol{x}_\theta(\boldsymbol{x}_t, t) + \sigma_t\Big(\frac{d\log\sigma_t}{dt} - \beta\Big)\boldsymbol{\epsilon}_\theta(\boldsymbol{x}_t, t).
\end{aligned}$$

Thus, we obtain the differential form of *dual prediction*. It is straightforward to verify that choosing $\beta = \frac{d}{dt}\log\alpha_t$ recovers the noise-prediction form, $\beta = \frac{d}{dt}\log\sigma_t$ recovers the data-prediction form, and $\beta = 0$ recovers the velocity-prediction form in Table 1.

**Integral Form of Dual Prediction.**   We next apply the variation-of-constants method to Eq. 11 to obtain the following integral representation:

$$\begin{aligned}
\boldsymbol{x}_{t_{i+1}} = {}& \exp\Big(\int_{t_i}^{t_{i+1}} \beta_u\,du\Big)\boldsymbol{x}_{t_i} + \\
& \int_{t_i}^{t_{i+1}} \exp\Big(\int_s^{t_{i+1}} \beta_u\,du\Big)\Big[\alpha_s\Big(\frac{d\log\alpha_s}{ds} - \beta_s\Big)\boldsymbol{x}_\theta\ +\ \sigma_s\Big(\frac{d\log\sigma_s}{ds} - \beta_s\Big)\boldsymbol{\epsilon}_\theta\Big]\,ds.
\end{aligned}$$

We reparameterize $\beta$ in terms of a new variable $\gamma \in \mathbb{R}$ as follows:

$$
\beta(\gamma) = \begin{cases} \gamma \dfrac{d \log \sigma_t}{dt} = \gamma \dfrac{\dot{\sigma}_t}{\sigma_t}, & \gamma \geq 0, \\ -\gamma \dfrac{d \log \alpha_t}{dt} = -\gamma \dfrac{\dot{\alpha}_t}{\alpha_t}, & \gamma < 0. \end{cases}
$$

Then we obtain the $\gamma$-interpolated integral form of dual prediction (Eq. 5).

## B.2 DERIVATION OF SAMPLING EQUATIONS

### B.2.1 DERIVATION OF FIRST-ORDER PREDICTOR

First, we take a first-order approximation of $\boldsymbol{x}_\theta(u)$ and $\boldsymbol{\epsilon}_\theta(v)$ in Eq. 7.

$$
\boldsymbol{x}_{t_{i+1}} \approx A\boldsymbol{x}_{t_i} + B\left[\boldsymbol{x}_\theta(u_i)\Delta L^{-1}(u_i) + \boldsymbol{\epsilon}_\theta(v_i)\Delta L^{-1}(v_i)\right].
$$

Here, we write $\boldsymbol{x}_\theta(u_i) := \boldsymbol{x}_\theta(\boldsymbol{x}_{u^{-1}(u_i)}, u^{-1}(u_i))$, and $\Delta L^{-1}(u_i) := L^{-1}(u_{i+1}) - L^{-1}(u_i)$; the definitions for $\boldsymbol{\epsilon}_\theta(v_i)$ and $\Delta L^{-1}(v_i)$ are analogous.

Next, while preserving first-order accuracy, we incorporate the $K(\Delta u_i; \kappa_u) = \mathcal{O}((\Delta u_i)^2)$ and $K(\Delta v_i; \kappa_v) = \mathcal{O}((\Delta v_i)^2)$, which yields the first-order predictor of Dual-Solver (Eq. 9).

### B.2.2 DERIVATION OF SECOND-ORDER PREDICTOR

First, we approximate $\boldsymbol{x}_\theta(u)$ and $\boldsymbol{\epsilon}_\theta(v)$ at $u_i$ and $v_i$ by a second-order backward-difference expansion.

$$
\boldsymbol{x}_{t_{i+1}} \approx A\boldsymbol{x}_{t_i} + B\left[\boldsymbol{x}_\theta(u_i)\Delta L^{-1}(u_i) + \frac{\Delta \boldsymbol{x}_\theta(u_{i-1})}{\Delta u_{i-1}}\int_{u_i}^{u_{i+1}}(u-u_i)\frac{dL^{-1}(u)}{du}du + \right.
$$
$$
\left. \boldsymbol{\epsilon}_\theta(v_i)\Delta L^{-1}(v_i) + \frac{\Delta \boldsymbol{\epsilon}_\theta(v_{i-1})}{\Delta v_{i-1}}\int_{v_i}^{v_{i+1}}(v-v_i)\frac{dL^{-1}(v)}{dv}dv\right].
$$

Using

$$
\frac{1}{\Delta u_{i-1}}\int_{u_i}^{u_{i+1}}(u-u_i)\frac{dL^{-1}(u)}{du}\,du
$$
$$
= \frac{1}{2r_i^{(u)}}\left(\left.\frac{dL^{-1}(u)}{du}\right|_{u_i}\Delta u_i + O((\Delta u_i)^2)\right)
$$
$$
= \frac{1}{2r_i^{(u)}}\left(\Delta L^{-1}(u_i) + \mathcal{O}((\Delta u_i)^2)\right)
$$

where $r_i^{(u)} := \frac{\Delta u_{i-1}}{\Delta u_i}$ and $r_i^{(v)} := \frac{\Delta v_{i-1}}{\Delta v_i}$, we obtain

$$
\boldsymbol{x}_{t_{i+1}} = A\boldsymbol{x}_{t_i} + B\left[\boldsymbol{x}_\theta(u_i)\Delta L^{-1}(u_i) + \frac{\Delta \boldsymbol{x}_\theta(u_{i-1})}{2r_i^{(u)}}\left(\Delta L^{-1}(u_i) + \mathcal{O}((\Delta u_i)^2)\right) + \right.
$$
$$
\left. \boldsymbol{\epsilon}_\theta(v_i)\Delta L^{-1}(v_i) + \frac{\Delta \boldsymbol{\epsilon}_\theta(v_{i-1})}{2r_i^{(v)}}\left(\Delta L^{-1}(v_i) + \mathcal{O}((\Delta v_i)^2)\right)\right].
$$

Next, while preserving second-order accuracy, we incorporate the $K(\Delta u_i; \kappa_u) = \mathcal{O}((\Delta u_i)^2)$ and $K(\Delta v_i; \kappa_v) = \mathcal{O}((\Delta v_i)^2)$ residual terms, which yields the following second-order predictor of Dual-Solver.

$$
\boldsymbol{x}_{t_{i+1}}^{\text{2nd-pred.}} = A\boldsymbol{x}_{t_i} + B\left[\boldsymbol{x}_\theta(u_i)\Delta L^{-1}(u_i) + \frac{\Delta \boldsymbol{x}_\theta(u_{i-1})}{2r_i^{(u)}}\left(\Delta L^{-1}(u_i) + K(\Delta u_i; \kappa_u)\right) \right.
$$
$$
\left. + \boldsymbol{\epsilon}_\theta(v_i)\Delta L^{-1}(v_i) + \frac{\Delta \boldsymbol{\epsilon}_\theta(v_{i-1})}{2r_i^{(v)}}\left(\Delta L^{-1}(v_i) + K(\Delta v_i; \kappa_v)\right)\right]
$$

$$(12)$$

### B.2.3 Derivation of Second-Order Corrector

First, we approximate $\boldsymbol{x}_\theta(u)$ and $\boldsymbol{\epsilon}_\theta(v)$ at $u_i$ and $v_i$ by a second-order forward-difference expansion.

$$\boldsymbol{x}_{t_{i+1}} \approx A\boldsymbol{x}_{t_i} + B\left[\boldsymbol{x}_\theta(u_i)\Delta L^{-1}(u_i) + \frac{\Delta\boldsymbol{x}_\theta(u_i)}{\Delta u_i}\int_{u_i}^{u_{i+1}}(u-u_i)\frac{dL^{-1}(u)}{du}du + \right.$$
$$\left. \boldsymbol{\epsilon}_\theta(v_i)\Delta L^{-1}(v_i) + \frac{\Delta\boldsymbol{\epsilon}_\theta(v_i)}{\Delta v_i}\int_{v_i}^{v_{i+1}}(v-v_i)\frac{dL^{-1}(v)}{dv}dv\right].$$

Using

$$\frac{1}{\Delta u_i}\int_{u_i}^{u_{i+1}}(u-u_i)\frac{dL^{-1}(u)}{du}\,du$$
$$= \frac{1}{2}\left(\left.\frac{dL^{-1}(u)}{du}\right|_{u_i}\Delta u_i + O((\Delta u_i)^2)\right)$$
$$= \frac{1}{2}\left(\Delta L^{-1}(u_i) + \mathcal{O}((\Delta u_i)^2)\right)$$

we obtain

$$\boldsymbol{x}_{t_{i+1}} = A\boldsymbol{x}_{t_i} + B\left[\boldsymbol{x}_\theta(u_i)\Delta L^{-1}(u_i) + \frac{\Delta\boldsymbol{x}_\theta(u_i)}{2}\left(\Delta L^{-1}(u_i) + \mathcal{O}((\Delta u_i)^2)\right)\right.$$
$$\left. + \boldsymbol{\epsilon}_\theta(v_i)\Delta L^{-1}(v_i) + \frac{\Delta\boldsymbol{\epsilon}_\theta(v_i)}{2}\left(\Delta L^{-1}(v_i) + \mathcal{O}((\Delta v_i)^2)\right)\right].$$

Next, while preserving second-order accuracy, we incorporate the $K(\Delta u_i; \kappa_u) = \mathcal{O}((\Delta u_i)^2)$ and $K(\Delta v_i; \kappa_v) = \mathcal{O}((\Delta v_i)^2)$ residual terms, which yields the second-order corrector of Dual-Solver (Eq. 10).

## C   Local Truncation Error

### C.1   Local Truncation Error of First-Order Predictor

**Theorem C.1.** *Assume that $\boldsymbol{x}_\theta(u)$ and $\boldsymbol{\epsilon}_\theta(v)$ are $C^1$ on $[u_i, u_{i+1}]$ and $[v_i, v_{i+1}]$, respectively. Let $\boldsymbol{x}_{t_{i+1}}^{exact}$ denote the exact update in equation 7, and let $\boldsymbol{x}_{t_{i+1}}^{1st\text{-}pred.}$ denote the first–order predictor defined in equation 9. Then we have*

$$\left\|\boldsymbol{x}_{t_{i+1}}^{exact} - \boldsymbol{x}_{t_{i+1}}^{1st\text{-}pred.}\right\| = \mathcal{O}\big((\Delta u_i)^2 + (\Delta v_i)^2\big).$$

*Proof.* From Eq. 7, the exact update can be written as

$$\boldsymbol{x}_{t_{i+1}}^{\text{exact}} = A\boldsymbol{x}_{t_i} + B\left[I_u + I_v\right],$$

where $I_u = \int_{u_i}^{u_{i+1}}\frac{dL^{-1}(u)}{du}\,\boldsymbol{x}_\theta(u)\,du$ and $I_v = \int_{v_i}^{v_{i+1}}\frac{dL^{-1}(v)}{dv}\,\boldsymbol{\epsilon}_\theta(v)\,dv$.

Eq. 9 defines the first-order predictor

$$\boldsymbol{x}_{t_{i+1}}^{\text{1st-pred.}} = A\boldsymbol{x}_{t_i} + B\left[I_u' + I_v'\right],$$

with $I_u' = \boldsymbol{x}_\theta(u_i)\left(\Delta L^{-1}(u_i) + K(\Delta u_i; \kappa_u)\right)$ and $I_v' = \boldsymbol{\epsilon}_\theta(v_i)\left(\Delta L^{-1}(v_i) + K(\Delta v_i; \kappa_v)\right)$.

The accuracy can be obtained by taking a Taylor expansion of $I_u$ and estimating the order of $I_u - I_u'$.

**Expansion of $I_u$**

$$I_u = \int_{u_i}^{u_{i+1}}\frac{dL^{-1}(u)}{du}\boldsymbol{x}_\theta(u)du$$
$$= \boldsymbol{x}_\theta(u_i)\Delta L^{-1}(u_i) + \mathcal{O}((\Delta u_i)^2).$$

**Order estimate of $I_u - I'_u$**
$$I_u - I'_u = \mathcal{O}\big((\Delta u_i)^2\big) \; - \; \boldsymbol{x}_\theta(u_i)\, K(\Delta u_i; \kappa_u).$$

Since $\boldsymbol{x}_\theta(u_i) = \mathcal{O}(1)$ and $K(\Delta u_i; \kappa_u) = \mathcal{O}((\Delta u_i)^2)$, it follows that

$$\boldsymbol{x}_\theta(u_i)\, K(\Delta u_i; \kappa_u) = \mathcal{O}\big((\Delta u_i)^2\big).$$

Therefore,

$$I_u - I'_u = \mathcal{O}\big((\Delta u_i)^2\big).$$

The bound $I_v - I'_v = \mathcal{O}\big((\Delta v_i)^2\big)$ can be derived in the same way.

**Conclusion**
$$
\begin{aligned}
\boldsymbol{x}^{\text{exact}}_{t_{i+1}} - \boldsymbol{x}^{\text{1st-pred.}}_{t_{i+1}} &= B\big[(I_u - I'_u) + (I_v - I'_v)\big] \\
&= B\Big[\mathcal{O}((\Delta u_i)^2) + \mathcal{O}((\Delta v_i)^2)\Big] \\
&= \mathcal{O}\big((\Delta u_i)^2 + (\Delta v_i)^2\big).
\end{aligned}
$$

$\square$

## C.2 Local Truncation Error of Second-Order Predictor

**Theorem C.2.** *Assume that $\boldsymbol{x}_\theta(u)$ and $\boldsymbol{\epsilon}_\theta(v)$ are $C^2$ on $[u_{i-1}, u_{i+1}]$ and $[v_{i-1}, v_{i+1}]$, respectively. Let $\boldsymbol{x}^{exact}_{t_{i+1}}$ denote the exact update in equation 7, and let $\boldsymbol{x}^{2nd\text{-}pred.}_{t_{i+1}}$ denote the second–order predictor defined in equation 12. Then we have*

$$\big\|\boldsymbol{x}^{exact}_{t_{i+1}} - \boldsymbol{x}^{2nd\text{-}pred.}_{t_{i+1}}\big\| \;=\; \mathcal{O}\big((\Delta u_i)^3 + (\Delta v_i)^3\big).$$

*Proof.* From Eq. 7, the exact update is given by

$$\boldsymbol{x}^{\text{exact}}_{t_{i+1}} = A\boldsymbol{x}_{t_i} \; + \; B\left[I_u \; + \; I_v\right],$$

where $I_u = \int_{u_i}^{u_{i+1}} \frac{dL^{-1}(u)}{du}\, \boldsymbol{x}_\theta(u)\, du$ and $I_v = \int_{v_i}^{v_{i+1}} \frac{dL^{-1}(v)}{dv}\, \boldsymbol{\epsilon}_\theta(v)\, dv$.

Eq. 12 defines the second-order predictor

$$\boldsymbol{x}^{\text{2nd-pred.}}_{t_{i+1}} = A\boldsymbol{x}_{t_i} \; + \; B\left[I'_u \; + \; I'_v\right],$$

with $I'_u = \boldsymbol{x}_\theta(u_i)\, \Delta L^{-1}(u_i) + \frac{\Delta \boldsymbol{x}_\theta(u_{i-1})}{2 r^{(u)}_i}\big(\Delta L^{-1}(u_i) + K(\Delta u_i; \kappa_u)\big)$ and $I'_v = \boldsymbol{\epsilon}_\theta(v_i)\, \Delta L^{-1}(v_i) + \frac{\Delta \boldsymbol{\epsilon}_\theta(v_{i-1})}{2 r^{(v)}_i}\big(\Delta L^{-1}(v_i) + K(\Delta v_i; \kappa_v)\big)$.

The accuracy can be obtained by taking a Taylor expansion of $I_u$ and estimating the order of $I_u - I'_u$.

**Expansion of $I_u$**
$$
\begin{aligned}
I_u &= \int_{u_i}^{u_{i+1}} \frac{dL^{-1}(u)}{du}\, \boldsymbol{x}_\theta(u)\, du \\
&= \boldsymbol{x}_\theta(u_i)\, \Delta L^{-1}(u_i) + \int_{u_i}^{u_{i+1}} \frac{dL^{-1}(u)}{du}\big(\boldsymbol{x}_\theta(u) - \boldsymbol{x}_\theta(u_i)\big)\, du \\
&= \boldsymbol{x}_\theta(u_i)\, \Delta L^{-1}(u_i) + \boldsymbol{x}'_\theta(u_i)\int_{u_i}^{u_{i+1}} (u - u_i)\frac{dL^{-1}(u)}{du}\, du \; + \; \mathcal{O}\big((\Delta u_i)^3\big) \\
&= \boldsymbol{x}_\theta(u_i)\, \Delta L^{-1}(u_i) + \frac{1}{2}\, \boldsymbol{x}'_\theta(u_i)\, \Delta u_i\, \Delta L^{-1}(u_i) \; + \; \mathcal{O}\big((\Delta u_i)^3\big) \\
&= \boldsymbol{x}_\theta(u_i)\, \Delta L^{-1}(u_i) + \frac{1}{2 r^{(u)}_i}\, \Delta \boldsymbol{x}_\theta(u_{i-1})\, \Delta L^{-1}(u_i) \; + \; \mathcal{O}\big((\Delta u_i)^3\big).
\end{aligned}
$$

**Order estimate of $I_u - I'_u$**

$$I_u - I'_u = \mathcal{O}\big((\Delta u_i)^3\big) \;-\; \frac{1}{2r_i^{(u)}} \, \Delta \boldsymbol{x}_\theta(u_{i-1}) \, K(\Delta u_i; \kappa_u).$$

Since $\Delta \boldsymbol{x}_\theta(u_{i-1}) = \mathcal{O}(\Delta u_i)$ and $K(\Delta u_i; \kappa_u) = \mathcal{O}((\Delta u_i)^2)$, it follows that

$$\Delta \boldsymbol{x}_\theta(u_{i-1}) \, K(\Delta u_i; \kappa_u) = \mathcal{O}\big((\Delta u_i)^3\big).$$

Therefore,

$$I_u - I'_u = \mathcal{O}\big((\Delta u_i)^3\big).$$

The bound $I_v - I'_v = \mathcal{O}\big((\Delta v_i)^3\big)$ can be derived in the same way.

**Conclusion**

$$
\begin{aligned}
\boldsymbol{x}_{t_{i+1}}^{\text{exact}} - \boldsymbol{x}_{t_{i+1}}^{\text{2nd-pred.}} &= B\big[(I_u - I'_u) + (I_v - I'_v)\big] \\
&= B\Big[\mathcal{O}((\Delta u_i)^3) + \mathcal{O}((\Delta v_i)^3)\Big] \\
&= \mathcal{O}\big((\Delta u_i)^3 + (\Delta v_i)^3\big).
\end{aligned}
$$

$\square$

## C.3   LOCAL TRUNCATION ERROR OF SECOND-ORDER CORRECTOR

**Theorem C.3.** *Assume that $\boldsymbol{x}_\theta(u)$ and $\boldsymbol{\epsilon}_\theta(v)$ are $C^2$ on $[u_i, u_{i+1}]$ and $[v_i, v_{i+1}]$, respectively. Let $\boldsymbol{x}_{t_{i+1}}^{exact}$ denote the exact update in equation 7, and let $\boldsymbol{x}_{t_{i+1}}^{2nd\text{-}corr.}$ denote the second–order corrector defined in equation 10. Then we have*

$$\big\| \boldsymbol{x}_{t_{i+1}}^{exact} - \boldsymbol{x}_{t_{i+1}}^{2nd\text{-}corr.} \big\| \;=\; \mathcal{O}\big((\Delta u_i)^3 + (\Delta v_i)^3\big).$$

*Proof.* From Eq. 7, the exact update is expressed as

$$\boldsymbol{x}_{t_{i+1}}^{\text{exact}} = A\boldsymbol{x}_{t_i} \;+\; B\left[I_u \;+\; I_v\right],$$

where $I_u = \int_{u_i}^{u_{i+1}} \frac{dL^{-1}(u)}{du} \, \boldsymbol{x}_\theta(u) \, du$ and $I_v = \int_{v_i}^{v_{i+1}} \frac{dL^{-1}(v)}{dv} \, \boldsymbol{\epsilon}_\theta(v) \, dv$.

Eq. 10 defines the second-order corrector

$$\boldsymbol{x}_{t_{i+1}}^{\text{2nd-corr.}} = A\boldsymbol{x}_{t_i} \;+\; B\left[I'_u \;+\; I'_v\right],$$

where $I'_u = \boldsymbol{x}_\theta(u_i) \, \Delta L^{-1}(u_i) + \frac{\Delta \boldsymbol{x}_\theta(u_i)}{2}\big(\Delta L^{-1}(u_i) + K(\Delta u_i; \kappa_u)\big)$ and $I'_v = \boldsymbol{\epsilon}_\theta(v_i) \, \Delta L^{-1}(v_i) + \frac{\Delta \boldsymbol{\epsilon}_\theta(v_i)}{2}\big(\Delta L^{-1}(v_i) + K(\Delta v_i; \kappa_v)\big)$.

The accuracy can be obtained by taking a Taylor expansion of $I_u$ and estimating the order of $I_u - I'_u$.

**Expansion of $I_u$**

$$
\begin{aligned}
I_u &= \int_{u_i}^{u_{i+1}} \frac{dL^{-1}(u)}{du} \boldsymbol{x}_\theta(u) du \\
&= \boldsymbol{x}_\theta(u_i)\Delta L^{-1}(u_i) + \int_{u_i}^{u_{i+1}} \frac{dL^{-1}(u)}{du}\big(\boldsymbol{x}_\theta(u) - \boldsymbol{x}_\theta(u_i)\big) du \\
&= \boldsymbol{x}_\theta(u_i)\Delta L^{-1}(u_i) + \boldsymbol{x}'_\theta(u_i)\int_{u_i}^{u_{i+1}}(u - u_i)\frac{dL^{-1}(u)}{du} du \;+\; \mathcal{O}\big((\Delta u_i)^3\big) \\
&= \boldsymbol{x}_\theta(u_i)\Delta L^{-1}(u_i) + \frac{1}{2}\boldsymbol{x}'_\theta(u_i)\Delta u_i \Delta L^{-1}(u_i) \;+\; \mathcal{O}\big((\Delta u_i)^3\big) \\
&= \boldsymbol{x}_\theta(u_i)\Delta L^{-1}(u_i) + \frac{1}{2}\Delta \boldsymbol{x}_\theta(u_i)\Delta L^{-1}(u_i) + \mathcal{O}\big((\Delta u_i)^3\big).
\end{aligned}
$$

**Order estimate of $I_u - I'_u$**

$$I_u - I'_u = \mathcal{O}\big((\Delta u_i)^3\big) \; - \; \frac{1}{2}\,\Delta\boldsymbol{x}_\theta(u_i)\,K(\Delta u_i;\kappa_u).$$

Since $\Delta\boldsymbol{x}_\theta(u_i) = \mathcal{O}(\Delta u_i)$ and $K(\Delta u_i;\kappa_u) = \mathcal{O}((\Delta u_i)^2)$, it follows that

$$\Delta\boldsymbol{x}_\theta(u_i)\,K(\Delta u_i;\kappa_u) = \mathcal{O}\big((\Delta u_i)^3\big).$$

Therefore,

$$I_u - I'_u = \mathcal{O}\big((\Delta u_i)^3\big).$$

The bound $I_v - I'_v = \mathcal{O}\big((\Delta v_i)^3\big)$ can be derived in the same way.

**Conclusion**

$$
\begin{aligned}
\boldsymbol{x}_{t_{i+1}}^{\text{exact}} - \boldsymbol{x}_{t_{i+1}}^{\text{2nd-corr.}} &= B\big[(I_u - I'_u) + (I_v - I'_v)\big] \\
&= B\Big[\mathcal{O}((\Delta u_i)^3) + \mathcal{O}((\Delta v_i)^3)\Big] \\
&= \mathcal{O}\big((\Delta u_i)^3 + (\Delta v_i)^3\big).
\end{aligned}
$$

$\square$

# D EXPERIMENTAL DETAILS

## D.1 SETUP

**Environment details.** We run all experiments on a single NVIDIA RTX 6000 pro (Driver 575.57.08) under Ubuntu 24.04 with Python 3.11.13, PyTorch 2.8.0, and CUDA 12.9.

**Backbone Details.** We evaluate DiT-XL/2 (Peebles & Xie, 2023), GM-DiT (Chen et al., 2025), SANA (Xie et al., 2024), and PixArt-$\alpha$ (Chen et al., 2023). All models are obtained via the diffusers library (von Platen et al., 2022) pipelines and run in evaluation mode with bfloat16. The model identifiers are:

- DiT: `facebook/DiT-XL-2-256`
- GM-DiT: `Lakonik/gmflow_imagenet_k8_ema`
- SANA: `Efficient-Large-Model/Sana_600M_512px_diffusers`
- PixArt-$\alpha$: `PixArt-alpha/PixArt-XL-2-512x512`

**Solver Details.** The solvers used in our experiments are the diffusion-dedicated DDIM (Song et al., 2021a), the second-order multistep DPM-Solver++ (Lu et al., 2022b), and the learned solvers BNS-Solver (Shaul et al., 2024) and DS-Solver (Wang et al., 2025). We use DDIM as the first-order counterpart of DPM-Solver++ (as proposed in Lu et al. (2022a)). Implementations of DPM-Solver++ and DS-Solver are taken from their official GitHub repositories[1,2], and BNS-Solver is implemented according to the paper.

**Learning Details.** We train with AdamW (Loshchilov & Hutter, 2017) using $\beta = (0.9, 0.999)$, $\epsilon = 1e-8$, and weight decay 0.01. The learning rate decays from $2 \times 10^{-3}$ to $1 \times 10^{-4}$ over 20k steps via cosine annealing (Loshchilov & Hutter, 2016). The batch size is fixed to 10 for all experiments. For regression-based learning, the teacher trajectory is a 200-step DDIM (Song et al., 2021a) method on 1k samples.

**Sampling Details.** We evaluate all backbones on an NFE grid of $3, 4, 5, 6, 7, 8, 9$. For classifier-free guidance (CFG; Ho & Salimans, 2021), we use backbone-specific fixed scales: DiT 1.5, GM-DiT 1.4, SANA 4.5, and PixArt-$\alpha$ 3.5. We compute FID using the publicly released ImageNet training-set statistics (Dhariwal & Nichol, 2021) for ImageNet, and MSCOCO 2014 validation-set statistics for text-to-image (SANA, PixArt-$\alpha$). CLIP scores are computed with the official RN101 variant (Radford et al., 2021). We generate 50k images for ImageNet and 30k images for text-to-image evaluation.

## D.2 QUANTITATIVE RESULTS

Table 6: **DiT** (Peebles & Xie, 2023): FID ($\downarrow$) vs. NFE. ImageNet generation, 50k samples.

| Method | 3 | 4 | 5 | 6 | 7 | 8 | 9 |
|---|---|---|---|---|---|---|---|
| DDIM (Song et al., 2021a) | 89.33 | 56.33 | 32.91 | 20.06 | 13.64 | 9.55 | 7.42 |
| DPM-Solver++(Lu et al., 2022b) | 88.46 | 47.64 | 22.19 | 11.49 | 7.06 | 5.19 | 4.43 |
| BNS-Solver(Shaul et al., 2024) | 103.26 | 38.20 | 14.53 | 6.37 | 4.25 | 3.37 | 3.05 |
| DS-Solver(Wang et al., 2025) | 67.31 | 17.31 | 7.66 | 5.46 | 3.79 | 3.08 | 3.02 |
| Dual-Solver (Ours) | **24.91** | **6.05** | **3.52** | **3.13** | **2.75** | **2.60** | **2.67** |

---

[1] `https://github.com/LuChengTHU/dpm-solver`
[2] `https://github.com/MCG-NJU/NeuralSolver`

Table 7: **GM-DiT** Chen et al. (2025): FID (↓) vs. NFE. ImageNet generation, 50k samples.

| Method | 3 | 4 | 5 | 6 | 7 | 8 | 9 |
|---|---|---|---|---|---|---|---|
| DDIM (Song et al., 2021a) | 70.15 | 35.98 | 19.70 | 12.55 | 9.03 | 7.01 | 5.78 |
| DPM-Solver++(Lu et al., 2022b) | 63.24 | 19.53 | 7.85 | 4.74 | 3.65 | 3.15 | 2.89 |
| BNS-Solver(Shaul et al., 2024) | 57.88 | 26.64 | 10.31 | 5.02 | 3.43 | 2.66 | 2.44 |
| DS-Solver(Wang et al., 2025) | 34.15 | 11.60 | 5.64 | 3.49 | **2.70** | 2.48 | 2.41 |
| Dual-Solver-R (Ours) | 45.53 | 14.43 | 7.49 | 3.75 | 2.77 | **2.44** | **2.32** |
| Dual-Solver-C (Ours) | **6.81** | **3.76** | **3.09** | **2.97** | 2.77 | 2.70 | 2.60 |

R = trajectory regression-based; C = classification-based.

Table 8: **PixArt-**$\alpha$ (Chen et al., 2023): FID (↓) and CLIP score (↑) vs. NFE. Text-to-image on MSCOCO 2014 (Lin et al., 2014) with 30k samples.

| Method | 3 | 4 | 5 | 6 | 7 | 8 | 9 |
|---|---|---|---|---|---|---|---|
| **FID (↓)** | | | | | | | |
| DDIM (Song et al., 2021a) | 71.37 | 45.21 | 35.12 | 30.92 | 28.60 | 27.39 | 26.58 |
| DPM-Solver++ (Lu et al., 2022b) | 76.01 | 41.77 | 31.48 | 27.83 | 26.56 | 25.82 | 25.48 |
| BNS-Solver (Shaul et al., 2024) | 125.65 | 66.94 | 41.31 | 32.55 | 28.55 | 25.18 | 24.15 |
| DS-Solver (Wang et al., 2025) | 118.01 | 66.09 | 43.62 | 29.74 | 28.25 | 26.57 | 25.22 |
| Dual-Solver (Ours) | **66.61** | **31.61** | **24.68** | **22.39** | **22.51** | **21.96** | **22.01** |
| **CLIP (RN101, ↑)** | | | | | | | |
| DDIM (Song et al., 2021a) | 0.4469 | 0.4670 | 0.4739 | 0.4764 | 0.4763 | 0.4778 | 0.4779 |
| DPM-Solver++ (Lu et al., 2022b) | 0.4422 | 0.4676 | 0.4746 | 0.4763 | 0.4768 | 0.4771 | 0.4771 |
| BNS-Solver (Shaul et al., 2024) | 0.4320 | 0.4582 | 0.4694 | 0.4733 | 0.4746 | 0.4755 | 0.4757 |
| DS-Solver (Wang et al., 2025) | 0.4303 | 0.4568 | 0.4692 | 0.4748 | 0.4754 | 0.4762 | 0.4764 |
| Dual-Solver (Ours) | **0.4499** | **0.4732** | **0.4784** | **0.4803** | **0.4806** | **0.4814** | **0.4815** |

Table 9: **SANA** (Xie et al., 2024): FID (↓) and CLIP score (RN101, ↑) vs. NFE. Text-to-image on MSCOCO 2014 (Lin et al., 2014) with 30k samples.

| Method | 3 | 4 | 5 | 6 | 7 | 8 | 9 |
|---|---|---|---|---|---|---|---|
| **FID (↓)** | | | | | | | |
| DDIM (Song et al., 2021a) | 45.05 | 27.72 | 23.93 | 23.06 | 22.99 | 22.96 | 22.97 |
| DPM-Solver++ (Lu et al., 2022b) | 45.33 | 26.12 | 22.56 | 22.48 | 22.79 | 22.90 | 23.11 |
| BNS-Solver (Shaul et al., 2024) | 48.16 | 26.37 | 21.04 | 20.66 | 20.79 | 21.13 | 21.64 |
| DS-Solver (Wang et al., 2025) | 48.65 | 29.15 | 21.66 | 20.65 | 21.43 | 21.80 | 22.27 |
| Dual-Solver (Ours) | **21.79** | **19.40** | **18.81** | **18.52** | **19.57** | **19.43** | **19.77** |
| **CLIP (↑)** | | | | | | | |
| DDIM (Song et al., 2021a) | 0.4656 | 0.4782 | 0.4813 | 0.4821 | 0.4824 | 0.4824 | 0.4824 |
| DPM-Solver++ (Lu et al., 2022b) | 0.4652 | 0.4779 | 0.4813 | 0.4821 | 0.4822 | 0.4820 | 0.4820 |
| BNS-Solver (Shaul et al., 2024) | 0.4651 | 0.4772 | 0.4800 | 0.4807 | 0.4808 | 0.4809 | 0.4808 |
| DS-Solver (Wang et al., 2025) | 0.4635 | 0.4765 | 0.4808 | 0.4816 | 0.4818 | 0.4818 | 0.4816 |
| Dual-Solver (Ours) | **0.4795** | **0.4801** | **0.4821** | **0.4837** | **0.4843** | **0.4844** | **0.4849** |

# E   CLASSIFIER ACCURACY VS. SAMPLE QUALITY

As examined in Sec. 6.2, we study the relationship between classifier accuracy and FID. In this section, Table 10 provides detailed numerical results, and Fig. 10 analyzes the trend from the perspectives of precision and recall (Kynkäänniemi et al., 2019). Table 10 reports the pretrained weights from TorchVision (maintainers & contributors, 2016) and the results of training Dual-Solver with each set of weights. Using a GM-DiT (Chen et al., 2025) backbone, we present FID, precision, and recall at NFE= 3 and 9. The best value for each metric is highlighted in bold.

Table 10: **ImageNet per-classifier metrics.** Top-5 accuracy, FID, precision/recall at NFE=3 and 9, and GFLOPs; FID is measured on 10k samples after training Dual-Solver for each classifier.

| Weights[3] | Top-5 Acc. (%) | FID@3 | FID@9 | Precision@3 | Precision@9 | Recall@3 | Recall@9 | GFLOPS |
|---|---|---|---|---|---|---|---|---|
| ViT_H_14_Weights.IMAGENET1K_SWAG_E2E_V1 | **98.694** | 10.71 | 6.72 | 0.8376 | 0.8960 | 0.7420 | 0.7311 | **1016.72** |
| RegNet_Y_128GF_Weights.IMAGENET1K_SWAG_LINEAR_V1 | 97.844 | 12.99 | 5.61 | 0.8230 | 0.8996 | 0.7327 | 0.7336 | 127.52 |
| RegNet_Y_16GF_Weights.IMAGENET1K_SWAG_LINEAR_V1 | 97.244 | 9.97 | 5.32 | **0.8573** | 0.9007 | 0.7392 | 0.7467 | 15.91 |
| ConvNeXt_Base_Weights.IMAGENET1K_V1 | 96.870 | 10.28 | 5.78 | 0.8529 | 0.9040 | 0.7359 | 0.7327 | 15.36 |
| EfficientNet_B5_Weights.IMAGENET1K_V1 | 96.628 | 11.53 | 6.01 | 0.8379 | 0.9048 | 0.7393 | 0.7310 | 10.27 |
| RegNet_Y_16GF_Weights.IMAGENET1K_V2 | 96.328 | 9.63 | 5.21 | 0.8559 | 0.9061 | 0.7396 | 0.7455 | 15.91 |
| SwinV2_T_Weights.IMAGENET1K_V1 | 96.132 | 9.73 | 5.34 | 0.8555 | 0.9078 | 0.7439 | 0.7410 | 5.94 |
| Swin_T_Weights.IMAGENET1K_V1 | 95.776 | 9.65 | 5.29 | 0.8559 | 0.9055 | 0.7448 | 0.7425 | 4.49 |
| RegNet_Y_32GF_Weights.IMAGENET1K_V1 | 95.340 | 9.55 | 5.04 | 0.8520 | 0.9043 | 0.7464 | 0.7455 | 32.28 |
| RegNet_Y_8GF_Weights.IMAGENET1K_V1 | 95.048 | 9.54 | 5.10 | 0.8559 | 0.9074 | 0.7438 | 0.7414 | 8.47 |
| RegNet_Y_3_2GF_Weights.IMAGENET1K_V1 | 94.576 | 9.50 | 5.01 | 0.8564 | 0.9065 | 0.7448 | 0.7436 | 3.18 |
| ResNet152_Weights.IMAGENET1K_V1 | 94.046 | 9.49 | 5.03 | 0.8537 | 0.9089 | 0.7493 | 0.7459 | 11.51 |
| ResNet101_Weights.IMAGENET1K_V1 | 93.546 | 9.59 | 4.98 | 0.8529 | 0.9067 | 0.7459 | 0.7454 | 7.80 |
| RegNet_Y_800MF_Weights.IMAGENET1K_V1 | 93.136 | **9.40** | 4.99 | 0.8532 | 0.9081 | **0.7508** | 0.7437 | 0.83 |
| MobileNet_V3_Large_Weights.IMAGENET1K_V2 | 92.566 | 9.44 | **4.87** | 0.8523 | 0.9062 | 0.7467 | 0.7475 | 0.22 |
| RegNet_Y_400MF_Weights.IMAGENET1K_V1 | 91.716 | 9.50 | **4.87** | 0.8560 | 0.9082 | 0.7470 | **0.7489** | 0.40 |
| RegNet_X_400MF_Weights.IMAGENET1K_V1 | 90.950 | 9.60 | 4.93 | 0.8553 | 0.9091 | 0.7448 | 0.7410 | 0.41 |
| MobileNet_V2_Weights.IMAGENET1K_V1 | 90.286 | 9.52 | 5.03 | 0.8541 | 0.9118 | 0.7471 | 0.7331 | 0.30 |
| ShuffleNet_V2_X1_0_Weights.IMAGENET1K_V1 | 88.316 | 11.57 | 5.51 | 0.8368 | 0.9017 | 0.7388 | 0.7436 | 0.14 |
| AlexNet_Weights.IMAGENET1K_V1 | 79.066 | 11.20 | 6.22 | 0.8300 | **0.9163** | 0.7455 | 0.7187 | 0.71 |

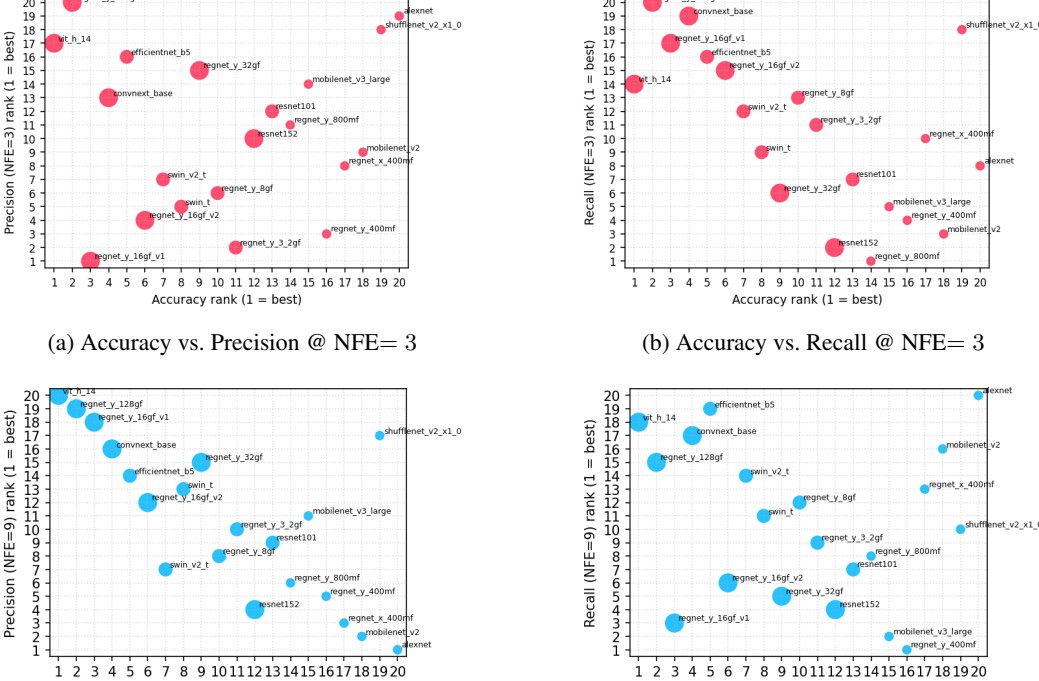

(a) Accuracy vs. Precision @ NFE= 3

(b) Accuracy vs. Recall @ NFE= 3

(c) Accuracy vs. Precision @ NFE= 9

(d) Accuracy vs. Recall @ NFE= 9

Figure 10: **Accuracy vs. precision/recall.** We select 20 TorchVision classifiers sorted by accuracy, learn the Dual-Solver for each, and report precision/recall at NFE=3 and 9 on 10k samples.

**Classifier accuracy vs. precision and recall.** Fig. 10 (a) and (b) plot precision and recall versus classifier accuracy at NFE= 3, respectively. The results show little relationship between accuracy

---

[3]https://docs.pytorch.org/vision/main/models.html

and precision, while accuracy versus recall follows the V-shape seen in Fig. 6. In other words, neither very high nor very low accuracy helps recall; a moderate level yields higher recall. The pattern differs at NFE= 9. Fig. 10 (c) and (d) plot precision and recall versus classifier accuracy at NFE= 9. Unlike the NFE= 3 case, precision exhibits a strong negative correlation with accuracy, and accuracy appears largely unrelated to recall.

**OpenCLIP accuracy vs. FID.**    Table 11 reports the FID evaluation used to select the CLIP model for learning Dual-Solver on the text-to-image task. All weights are from OpenCLIP (Ilharco et al., 2021) and are available from the official repository[4]. Using the SANA (Xie et al., 2024) backbone we trained the Dual-Solver for 20k steps. At NFE = 3 and 6, we generated 30k samples with MSCOCO 2014 (Lin et al., 2014) prompts and measured FID on the evaluation set. Based on these results, we chose the RN101 weights—which achieved an FID of 18.52 at NFE = 6—as the model for learning the Dual-Solver in the main text-to-image experiment. Notably, this result also indicates that models with somewhat lower classification accuracy can yield lower FID.

Table 11: **OpenCLIP per-classifier metrics.** MSCOCO accuracy, FID at NFE = 3 and 6, and GFLOPs; FID is measured on 30k samples after learning Dual-Solver for each classifier.

| Weights | MSCOCO Acc. (%) | FID@3 | FID@6 | GFLOPs |
|---|---|---|---|---|
| ViT-H-14-378-quickgelu, dfn5b | **63.76** | 23.98 | 23.28 | **1054.05** |
| coca_ViT-L-14, mscoco_finetuned_laion2b_s13b_b90k | 60.28 | 23.09 | 21.22 | 214.52 |
| EVA02-E-14, laion2b_s4b_b115k | 58.92 | 22.41 | 21.12 | 1007.93 |
| convnext_xxlarge, laion2b_s34b_b82k_augreg | 58.34 | 21.05 | 20.86 | 800.88 |
| ViT-B-16-SigLIP-256, webli | 57.24 | 21.03 | 19.87 | 57.84 |
| EVA02-L-14-336, merged2b_s6b_b61k | 56.05 | 23.15 | 23.28 | 167.50 |
| ViT-L-14, commonpool_xl_laion_s13b_b90k | 55.13 | 23.46 | 22.81 | 175.33 |
| convnext_base_w, laion_aesthetic_s13b_b82k | 52.38 | 20.97 | 19.86 | 49.38 |
| convnext_base_w_320, laion_aesthetic_s13b_b82k_augreg | 51.42 | **20.69** | 20.26 | 175.33 |
| ViT-B-16-plus-240, laion400m_e32 | 49.79 | 21.66 | 21.18 | 64.03 |
| ViT-B-32, laion2b_e16 | 47.68 | 23.75 | 23.49 | 14.78 |
| ViT-B-32-quickgelu, metaclip_fullcc | 46.62 | 22.46 | 21.42 | 14.78 |
| RN50x16, openai | 45.38 | 22.49 | 21.36 | 33.34 |
| ViT-B-32, laion400m_e31 | 43.27 | 22.40 | 21.70 | 14.78 |
| RN101, openai | 40.25 | 21.78 | **18.52** | 25.50 |
| ViT-B-16, commonpool_l_text_s1b_b8k | 37.30 | 23.50 | 23.54 | 41.09 |
| ViT-B-16, commonpool_l_s1b_b8k | 28.55 | 22.94 | 24.73 | 41.09 |
| ViT-B-32, commonpool_m_text_s128m_b4k | 14.52 | 22.39 | 22.55 | 14.78 |
| ViT-B-32, commonpool_s_clip_s13m_b4k | 2.24 | 22.32 | 21.31 | 14.78 |
| coca_ViT-B-32, mscoco_finetuned_laion2b_s13b_b90k | 0.60 | 23.76 | 23.14 | 33.34 |

---

[4]https://github.com/mlfoundations/open_clip

# F  LEARNED PARAMETERS

In Figs. 11, 12, 13, 14 we plot the Dual-Solver parameters learned via classification-based learning (Sec. 5.2), as defined in Sec. 4.2. For the DiT, GM-DiT, SANA, and PixArt-$\alpha$ backbones, we set CFG to 1.5, 1.4, 4.5, and 3.5, respectively, and train for 20k steps. We plot results at NFE = 3, 5, 7, and 9 for DiT and GM-DiT, and at NFE = 3, 4, 5, and 6 for SANA and PixArt-$\alpha$. Separate parameter sets are learned for the predictor and the corrector, and they are shown in different colors. Within the same backbone, the parameter curves are similar even across different NFEs. We conjecture that this arises from the backbone's intrinsic trajectory.

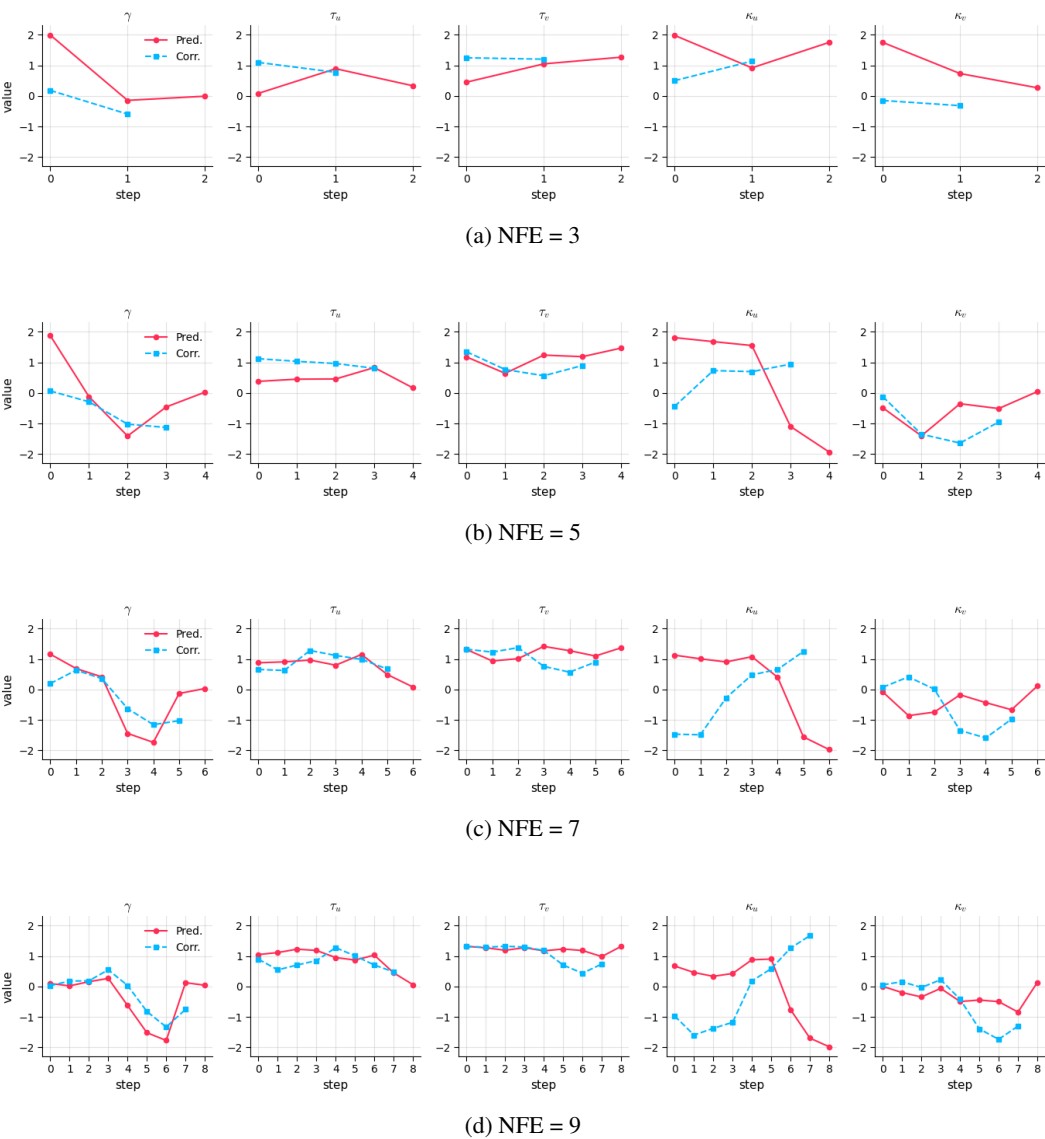

Figure 11: **Learned parameters.** $\{\gamma, \tau_u, \tau_v, \kappa_u, \kappa_v\}$ for DiT (Peebles & Xie, 2023).

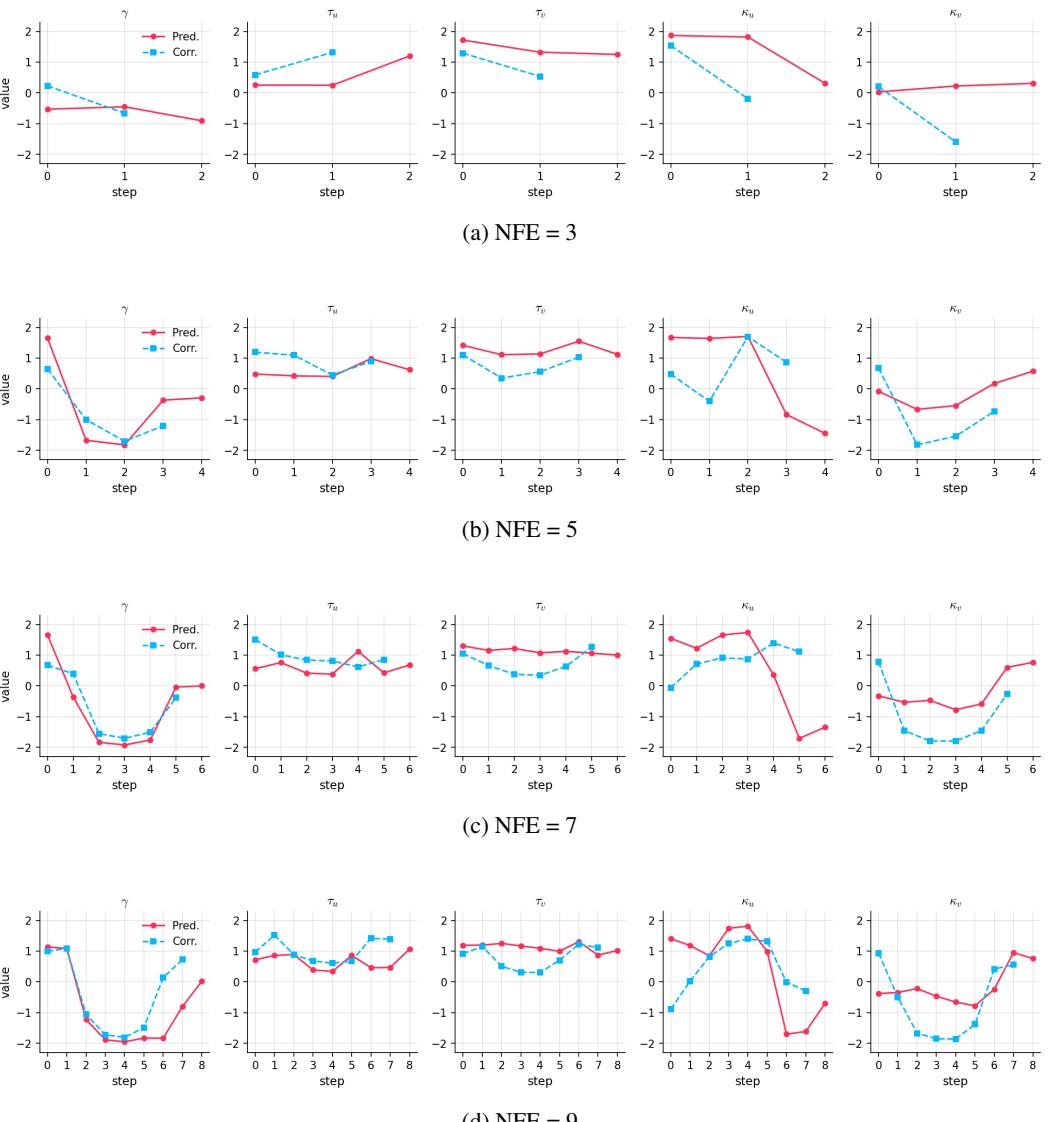

Figure 12: **Learned parameters.** $\{\gamma, \tau_u, \tau_v, \kappa_u, \kappa_v\}$ for GM-DiT (Chen et al., 2025).

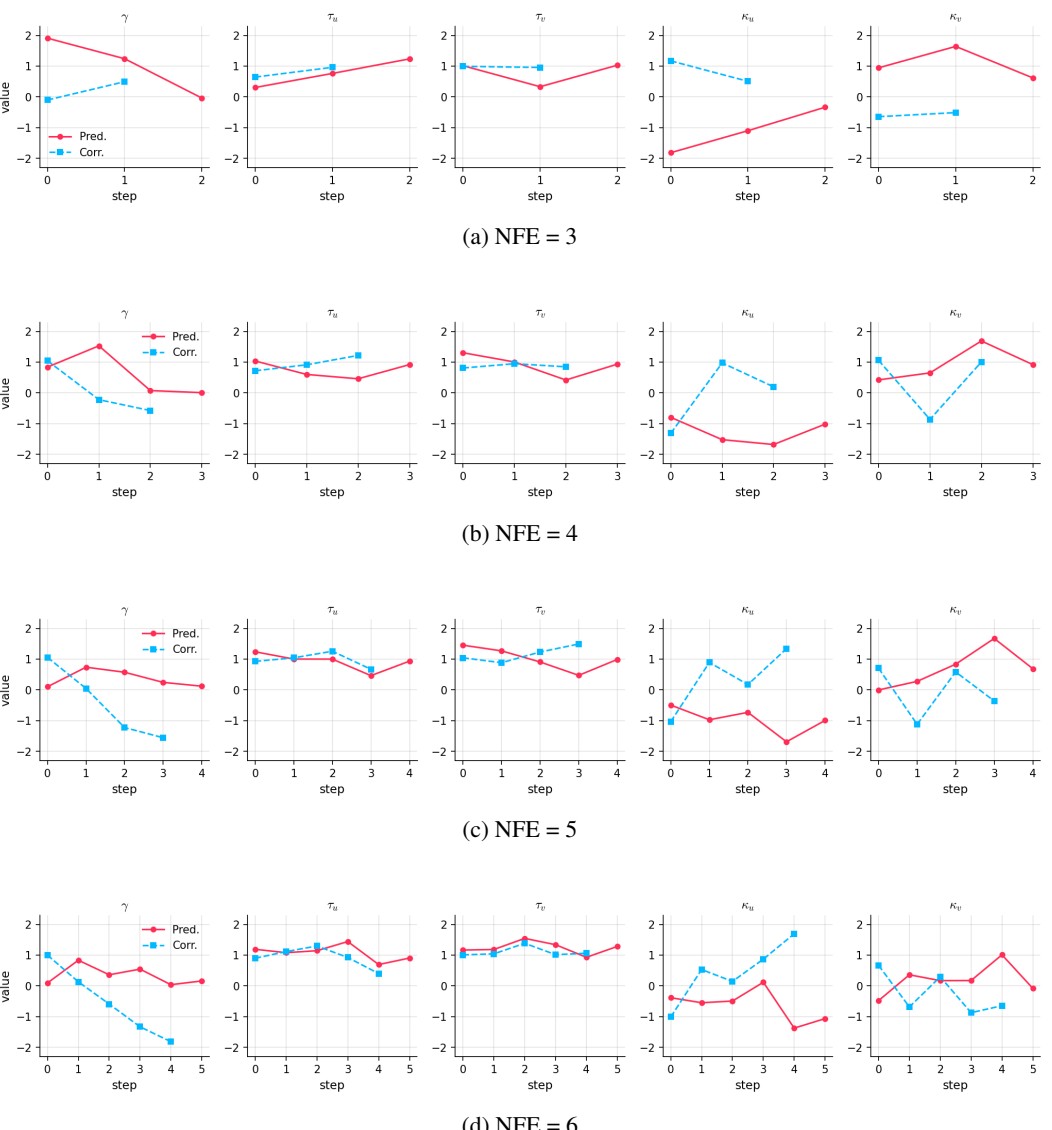

Figure 13: **Learned parameters.** $\{\gamma, \tau_u, \tau_v, \kappa_u, \kappa_v\}$ for SANA (Xie et al., 2024).

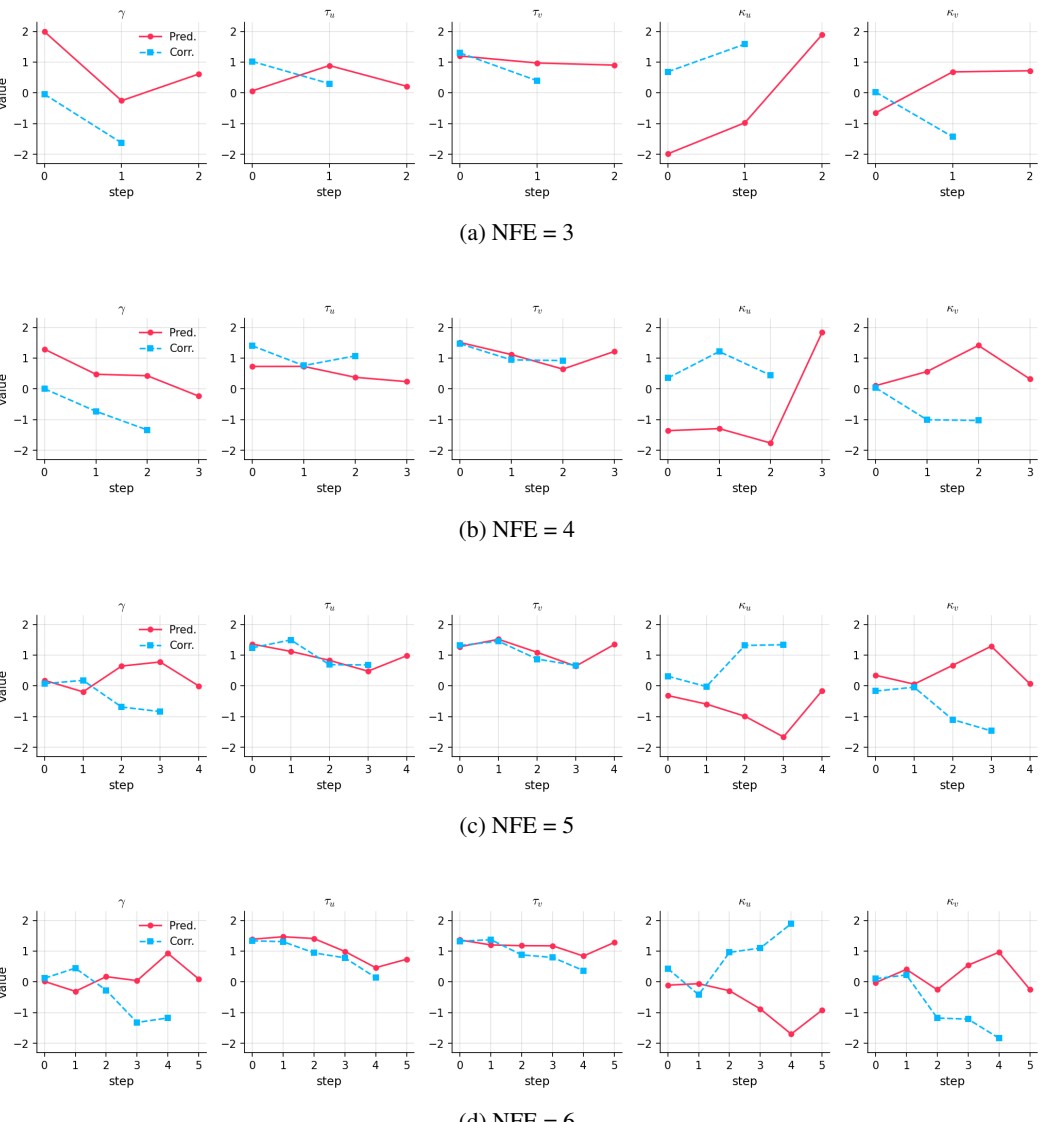

Figure 14: **Learned parameters.** $\{\gamma, \tau_u, \tau_v, \kappa_u, \kappa_v\}$ for PixArt-$\alpha$ (Chen et al., 2023).

## G  DETAILS OF PARAMETER INTERPOLATION ACROSS NFES

In this section, we describe how to interpolate the parameters discussed in Sec. 6.3 so that they can be used at other NFEs. The parameters of Dual-Solver,

$$\phi = \{\gamma^{\text{pred}}, \tau_u^{\text{pred}}, \tau_v^{\text{pred}}, \kappa_u^{\text{pred}}, \kappa_v^{\text{pred}}, \gamma^{\text{corr}}, \tau_u^{\text{corr}}, \tau_v^{\text{corr}}, \kappa_u^{\text{corr}}, \kappa_v^{\text{corr}}\},$$

are given as arrays whose length equals the NFE. (The corrector parameters have length NFE$-1$, and we match the length to NFE by repeating the last element.) For example, $\gamma^{\text{pred}} = (\gamma_0^{\text{pred}}, \ldots, \gamma_{NFE-1}^{\text{pred}})$. To interpolate these parameters, we consider the following linear interpolation scheme for a generic array.

**Definition G.1** (Linear interpolation). Let $f^{(M)} = (f_0^{(M)}, \ldots, f_{M-1}^{(M)})$ be an array of length $M$. The linearly interpolated array $\text{Interp}(f^{(M)}; N)$ of length $N$ is defined as follows. First, set

$$t_i = \frac{i}{N-1}(M-1), \qquad j_i = \lfloor t_i \rfloor, \qquad \alpha_i = t_i - j_i,$$

and for each $i = 0, \ldots, N-1$ define

$$\text{Interp}(f^{(M)}; N)[i] = (1 - \alpha_i)\, f_{j_i}^{(M)} + \alpha_i\, f_{j_i+1}^{(M)}.$$

**Definition G.2** (Averaged linear interpolation). Let $M < N < L$ be three NFEs, and let $f^{(M)} \in \mathbb{R}^M$ and $f^{(L)} \in \mathbb{R}^L$ be the corresponding arrays. We first obtain their linearly interpolated versions of length $N$,

$$\tilde{f}^{(M)} = \text{Interp}(f^{(M)}; N), \qquad \tilde{f}^{(L)} = \text{Interp}(f^{(L)}; N).$$

We then use the relative position of $N$ between $M$ and $L$ as weights and define

$$w_M = \frac{L-N}{L-M}, \qquad w_L = \frac{N-M}{L-M},$$

and, for each $i = 0, \ldots, N-1$,

$$\text{Interp}(f^{(M)}, f^{(L)}; N)[i] = w_M\, \tilde{f}^{(M)}[i] + w_L\, \tilde{f}^{(L)}[i].$$

Using this procedure, we obtain the array for an intermediate NFE from the two arrays at neighboring NFEs, and apply the same construction to every parameter in $\phi$. Examples of interpolated parameters are shown in Fig. 15. Specifically, we obtain the parameters for NFE = 4, 6, and 8 by interpolating those learned at NFE = (3, 5), (5, 7), and (7, 9), respectively.

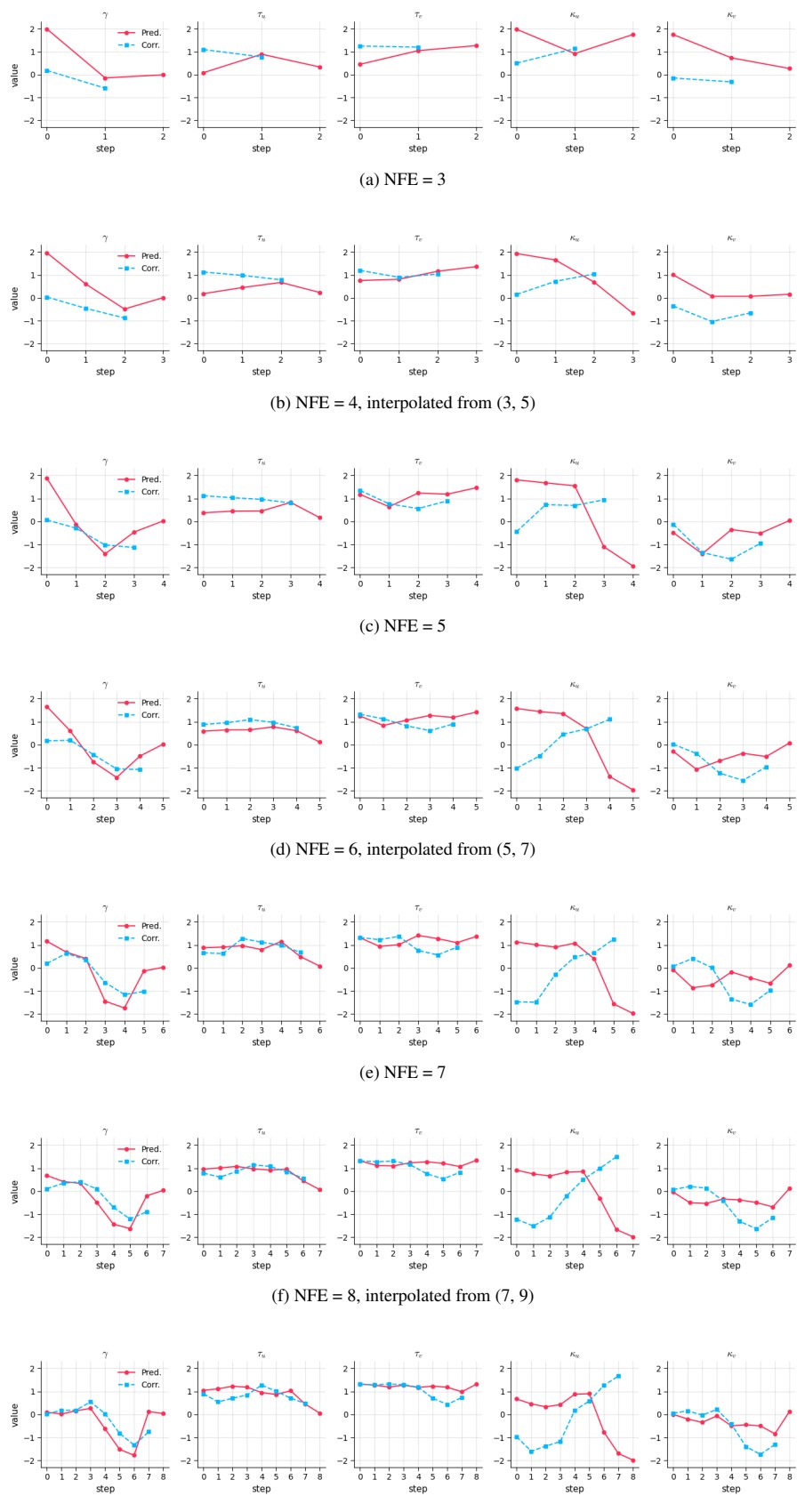

Figure 15: **Interpolated parameters.** $\{\gamma, \tau_u, \tau_v, \kappa_u, \kappa_v\}$ for DiT (Peebles & Xie, 2023).

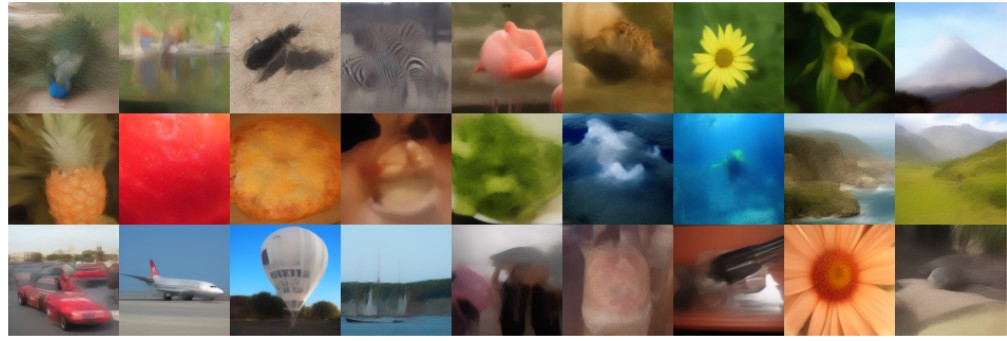

(a) DPM-Solver++ (Lu et al., 2022b)

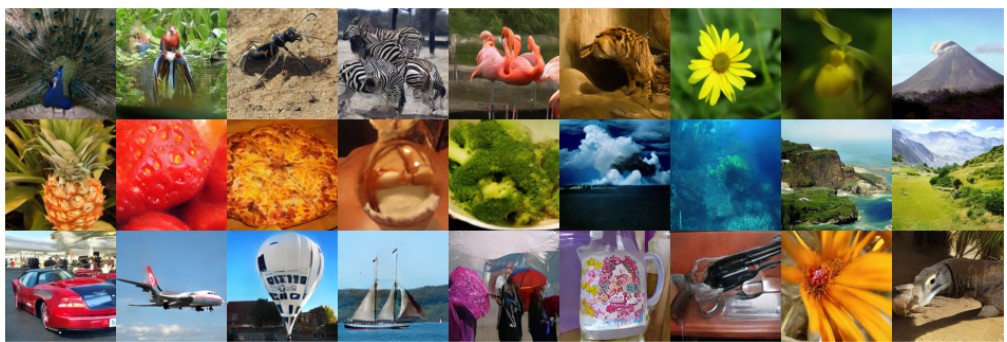

(b) Dual-Solver (Ours)

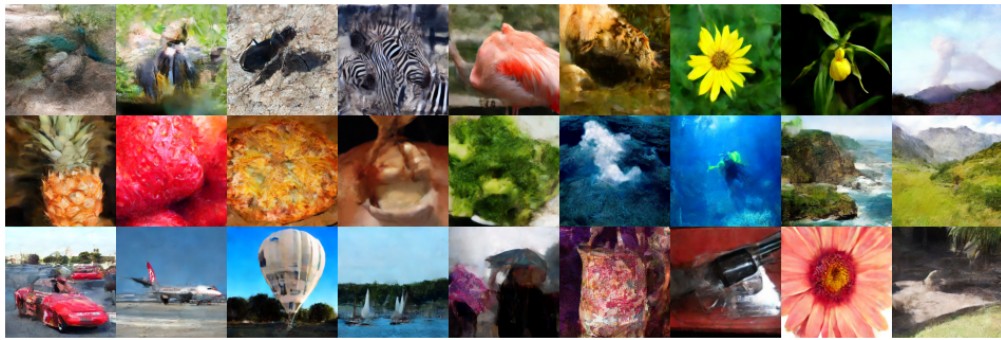

(c) BNS-Solver (Shaul et al., 2024)

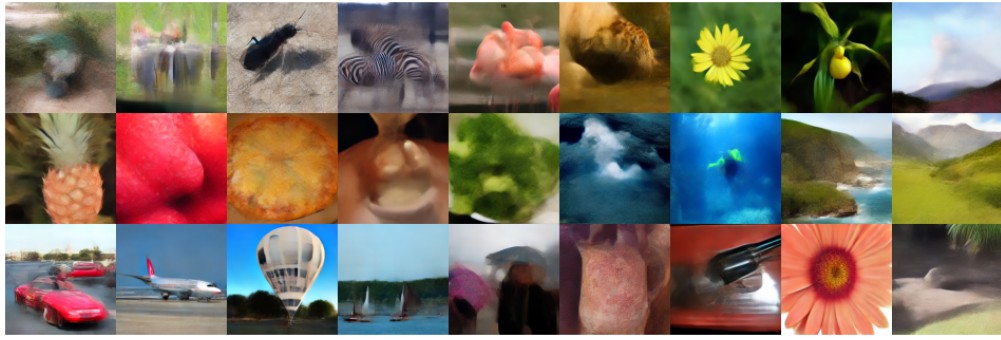

(d) DS-Solver (Wang et al., 2025)

Figure 16: **Additional sampling results.** DiT-XL/2 256×256 (NFE=4, CFG=1.5)

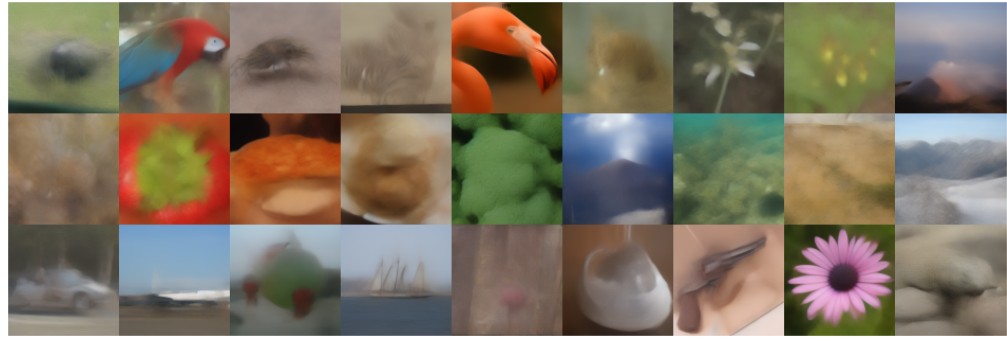

(a) DPM-Solver++ (Lu et al., 2022b)

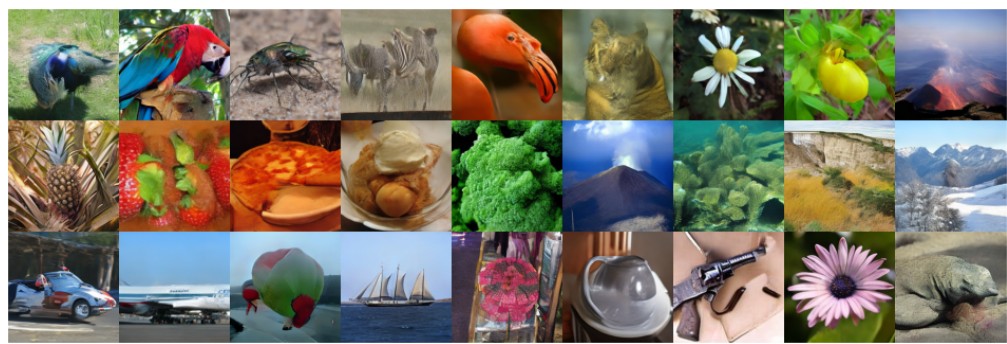

(b) Dual-Solver (Ours)

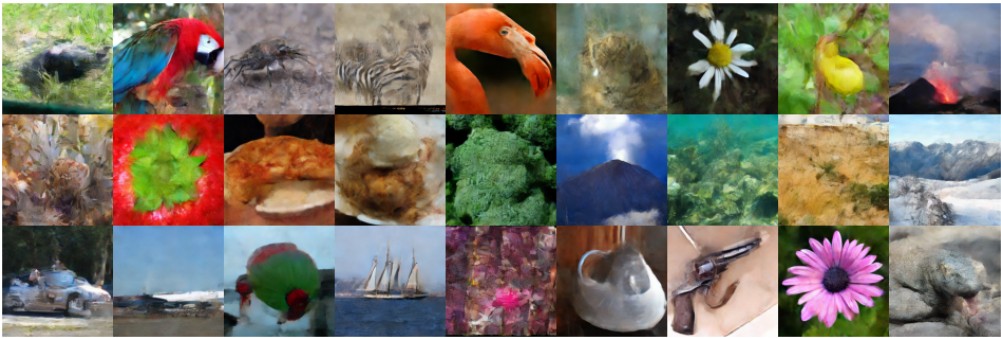

(c) BNS-Solver (Shaul et al., 2024)

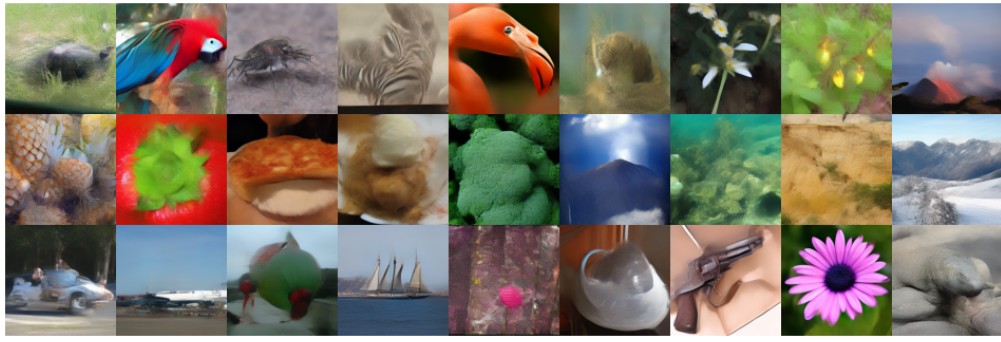

(d) DS-Solver (Wang et al., 2025)

Figure 17: **Additional sampling results.** GM-DiT 256×256 (NFE=3, CFG=1.4)

*Golden autumn park of falling leaves, graceful girl playing violin, flowing satin dress, impressionist brush strokes*

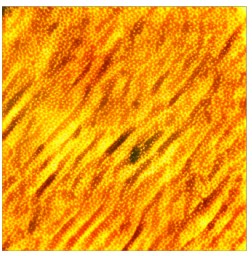 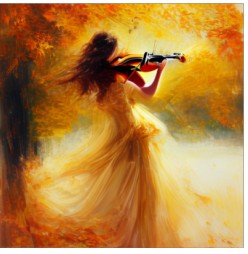 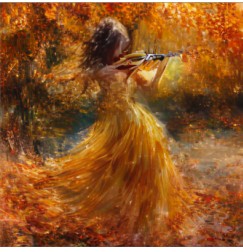 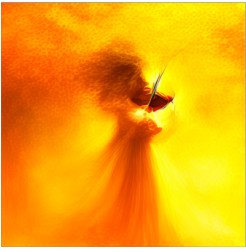

*Towering neon-lit cyberpunk skyline, armored samurai with glowing katana, chrome textures, cinematic digital art*

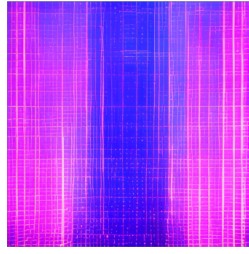 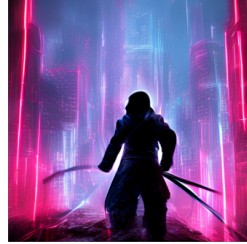 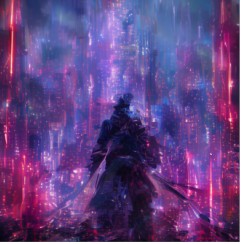 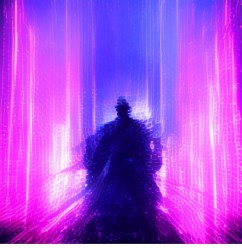

*Remote desert oasis at sunrise, powerful white stallion galloping, glossy mane and muscles, hyperreal photo realism*

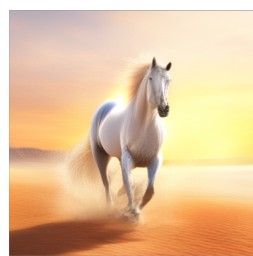 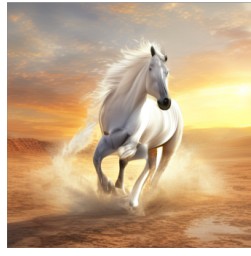 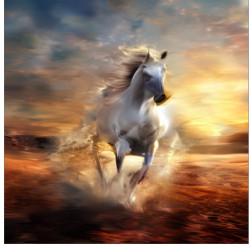 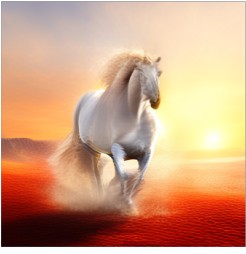

*Vibrant tropical beach at sunset, pirate ship anchored offshore, weathered wood hull, playful cartoon illustration*

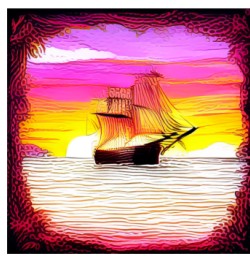 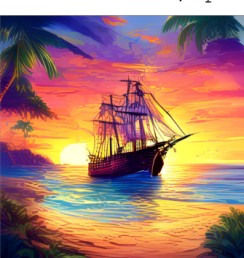 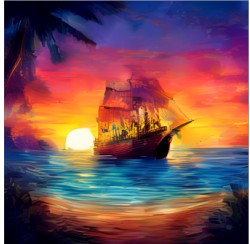 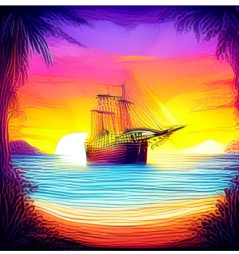

*Overgrown jungle temple ruins, spotted leopard stalking prey, sleek fur patterns, painterly realism concept art*

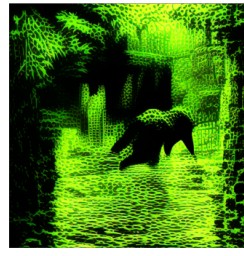 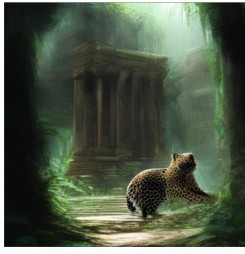 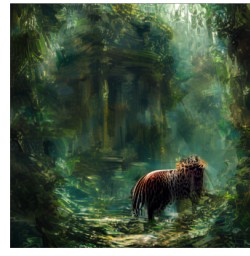 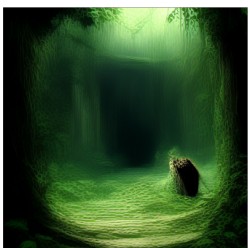

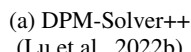

(a) DPM-Solver++ (Lu et al., 2022b)    (b) Dual-Solver (Ours)    (c) BNS-Solver (Shaul et al., 2024)    (d) DS-Solver (Wang et al., 2025)

Figure 18: **Additional sampling results.** SANA (Xie et al., 2024), NFE=3, CFG=4.5.

*In a misty emerald forest clearing, a majestic golden wolf with silky glowing fur, painted in luminous oil style*

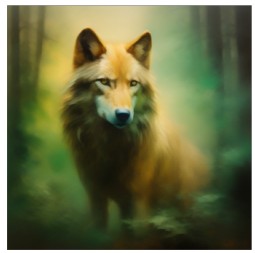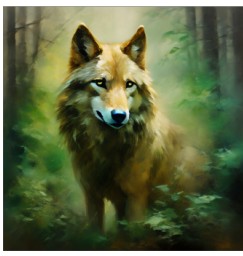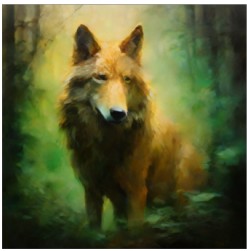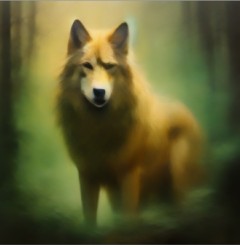

*Towering neon-lit cyberpunk skyline, armored samurai with glowing katana, chrome textures, cinematic digital art*

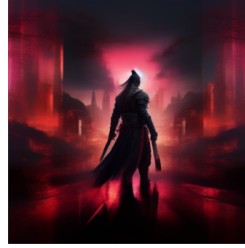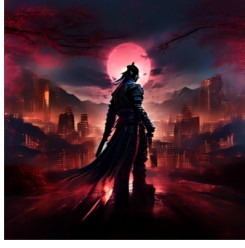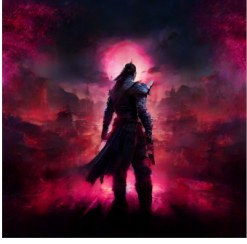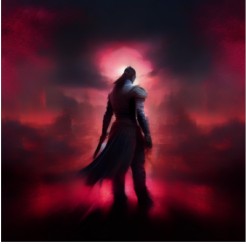

*Vast desert beneath a starry cosmos, colossal sphinx carved in sandstone, rough surface, surreal dreamlike painting*

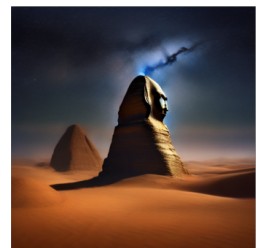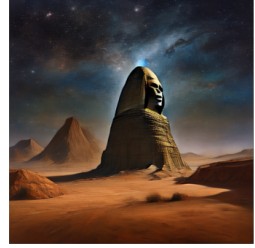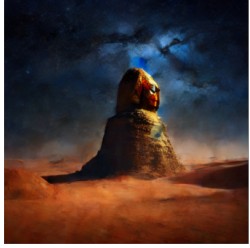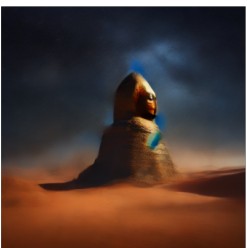

*Sleek futuristic laboratory interior, humanoid AI robot in motion, polished steel surfaces, sci-fi blueprint style*

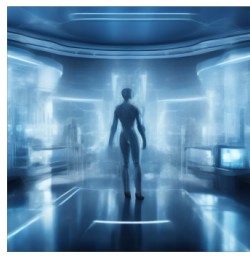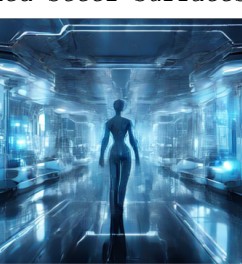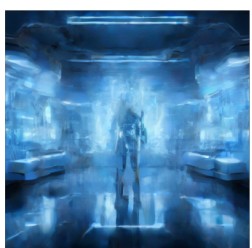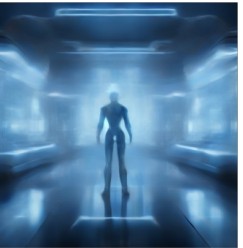

*Storm-drenched battlefield ruins, giant mech rising from fire, rusted armor plates, gritty photorealistic concept art*

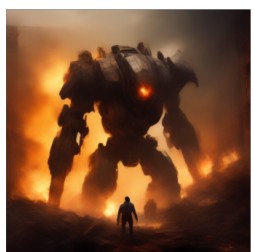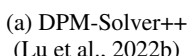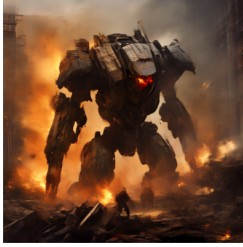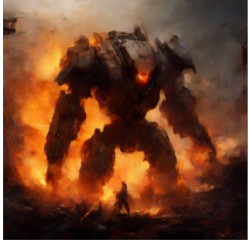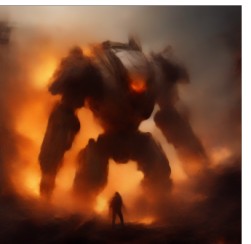

| (a) DPM-Solver++ | (b) Dual-Solver (Ours) | (c) BNS-Solver | (d) DS-Solver |
|---|---|---|---|
| (Lu et al., 2022b) | | (Shaul et al., 2024) | (Wang et al., 2025) |

Figure 19: **Additional sampling results.** PixArt-$\alpha$ (Chen et al., 2023), NFE=5, CFG=3.5.

