# OpenReview forum: "Dual-Solver: A Generalized ODE Solver for Diffusion Models with Dual Prediction"
_ICLR.cc/2026/Conference — ICLR 2026 Poster_

### Official Review · Reviewer_8gbp · 2025-10-21

**Soundness:** 4
**Presentation:** 4
**Contribution:** 3
**Rating:** 6
**Confidence:** 3

**Summary:**

Diffusion models deliver state-of-the-art image quality. However, sampling is costly at inference time because it requires many model evaluations (number of function evaluations, NFEs). To reduce NFEs, classical ODE multistep methods have been adopted. Yet differences in the choice of prediction type (noise/data/velocity) and integration domain (half log-SNR/noise-to-signal ratio) lead to different outcomes. This paper introduces Dual-Solver, which generalizes multistep samplers by introducing learnable parameters that continuously (i) interpolate among prediction types, (ii) select the integration domain, and (iii) adjust the residual terms. It maintains the traditional predictor-corrector structure and guarantees second-order local accuracy. These parameters are learned with a classification-based objective using a frozen pretrained classifier (e.g., ViT or CLIP). On ImageNet class-conditional generation (DiT, GM-DiT) and text-to-image (SANA, PixArtα), Dual-Solver consistently improves FID and CLIP scores in the low-NFE regime (NFE ≤ 6) across backbones.

**Strengths:**

1. Generalizes multistep samplers by introducing learnable parameters that continuously interpolate among different prediction types (noise/data/velocity), select the integration domain, and adjust residual terms, addressing the issue of inconsistent outcomes caused by differences in prediction type and integration domain choices.
2. Structural compatibility and guaranteed accuracy: Maintains the traditional predictor-corrector structure (ensuring compatibility with existing multistep method frameworks) while guaranteeing second-order local accuracy, balancing structural familiarity and precision.
3. Learns the key parameters via a classification-based objective, leveraging frozen pretrained classifiers (e.g., ViT, CLIP) — this avoids the need for heavy independent training and capitalizes on existing well-performing models.

**Weaknesses:**

see Questions

**Questions:**

1. For a single prediction model, does Dual-Solver need to first transform the prediction into eps and $x_0$?

2. Is dual prediction necessary? Given that a predefined noise scheduler has deterministic prediction transforms( eps to v, v to eps, v to x0), I have doubts regarding the arguments for dual prediction. Although other works [1, 2] use a single prediction, no discriminator is employed for learning.

[1] Shuai Wang, Zexian Li, Tianhui Song, Xubin Li, Tiezheng Ge, Bo Zheng, Limin Wang, et al. Differentiable solver search for fast diffusion sampling. arXiv preprint arXiv:2505.21114, 2025.

[2] Neta Shaul, Uriel Singer, Ricky TQ Chen, Matthew Le, Ali Thabet, Albert Pumarola, and Yaron Lipman. Bespoke non-stationary solvers for fast sampling of diffusion and flow models. arXiv preprint arXiv:2403.01329, 2024

---

> ### Author Response · Authors · 2025-11-21
> **Authors' response**
>
> We thank the reviewer for taking the time to review our paper. Below we provide responses to the raised questions.
>
> **For a single prediction model, does Dual-Solver need to first transform the prediction into eps and $\mathbf{x}_0$?**
>
> Dual-Solver performs sampling as follows: when using noise prediction or velocity prediction, it first obtains the corresponding data and noise predictions via the deterministic transform described in *Eq. 4* of the paper, and then applies the sampling equations. This transform uses only simple arithmetic operations and adds negligible computational overhead.
>
> **Is dual prediction necessary?**
>
> Dual-Solver introduces a learnable parameter $\gamma$ to interpolate among noise, velocity, and data predictions, thereby determining which prediction type is the best choice.
> As described in *Sec. 5 Solver Parameter Learning* of the paper, this decision is made through regression- or classification-based learning.
>
> We further conduct an ablation study in *Sec. 6.2* by fixing $\gamma$ to $1$, $0$, or $-1$ and measuring the resulting cross-entropy loss. The results show that making $\gamma$ learnable leads to significant performance improvements across all NFEs, with especially large gains in the low-NFE regime (NFE = 3, 5). These findings demonstrate that interpolating between prediction types via dual prediction is effective.

---

> > ### Comment · Reviewer_8gbp · 2025-11-23
> >
> > For a single prediction, you can convert it into a dual prediction model through simple parameterization, and then use a learnable coefficient to interpolate between the dual predictions. However, couldn’t the learnable coefficient and the conversion via simple parameterization be reparameterized into a single component? Furthermore, is there any necessity for such dual transform for a single prediction model.

---

> > > ### Author Response · Authors · 2025-11-25
> > >
> > > **Can the sampling equation of dual prediction be reparameterized using a single prediction?**
> > >
> > > As you pointed out, all sampling equations with dual prediction can indeed be reparameterized in terms of a single prediction, and we have added a new derivation in Appendix H to state this explicitly.
> > > For example, to obtain the first-order predictor sample $\mathbf x_{t_{i+1}}^{1st-pred.}$ in dual prediction, we need two among
> > > $\mathbf x_{t_{i}}$, $\mathbf x_\theta(\mathbf x_{t_i}, t_i)$, $\boldsymbol \epsilon_\theta(\mathbf x_{t_i}, t_i)$, and $\mathbf v_\theta(\mathbf x_{t_i}, t_i)$.
> > > Given any two of these four, the remaining two can be derived using Eq.(4) in the paper.
> > >
> > > Therefore, if $\mathbf x_{t_{i}}$ is given, a single prediction among $\mathbf x_\theta(\mathbf x_{t_i}, t_i)$, $\boldsymbol \epsilon_\theta(\mathbf x_{t_i}, t_i)$, or $\mathbf v_\theta(\mathbf x_{t_i}, t_i)$ is already sufficient.
> > > If the question “Is dual prediction necessary?” is interpreted as “Given $\mathbf x_{t_i}$, do the dual predictions $\mathbf x_\theta(\mathbf x_{t_i}, t_i), \boldsymbol \epsilon_\theta(\mathbf x_{t_i}, t_i)$ form a minimal set of quantities required to express the sampling equation?”, then the answer is “no”.
> > > One prediction is redundant with respect to the other.
> > >
> > > **Then why is dual prediction needed?**
> > >
> > > Our purpose in using dual prediction is not to simply increase the expressivity of the sampling equation by using $\mathbf x_\theta$ and $\boldsymbol \epsilon_\theta$ together.
> > > Given $\mathbf x_{t_i}$, one prediction is linearly dependent on the other, so, as you rightly pointed out, this does not by itself increase expressivity.
> > >
> > > Our goal is to interpolate, via the parameter $\gamma$, the discrepancy we observe between the noise sampling equation and the data sampling equation (Table 1) once they are discretized.
> > > The interpolation equation in integral form is given by Eq.(5), and its differential form is given as Eq.(9) in Appendix B.1. These integral and differential forms constitute the main contribution of our paper.
> > >
> > > To the best of our knowledge, there has been no prior work that interpolates between data and noise prediction (and furthermore also includes velocity prediction) using a single parameter. In the process of formulating the integral and differential forms, the two predictions $\mathbf x_\theta$ and $\boldsymbol\epsilon_\theta$ naturally appear simultaneously, which is why we refer to this as dual prediction. If the question is instead, “Is dual prediction necessary to derive the $\gamma$-parameterized integral form Eq.(5)?”, then the answer is “yes”.
> > >
> > > We thank the reviewer for the insightful questions, which helped us strengthen the paper and deepen the discussion. If there are any remaining issues that we have not fully addressed, we would be grateful for further comments.

---

### Official Review · Reviewer_qooq · 2025-10-26

**Soundness:** 3
**Presentation:** 2
**Contribution:** 2
**Rating:** 4
**Confidence:** 5

**Summary:**

This paper proposes Dual-Solver, a learnable, generalized ODE sampler that unifies noise/data/velocity predictions via dual prediction, and combines a log–linear domain transform with a second-order predictor–corrector and learnable residuals for stable few-step sampling.

It learns per-step parameters and the timestep schedule end-to-end using discriminative objectives (e.g., classifier/CLIP), yielding large quality gains at very low NFE across diverse backbones and tasks.

**Strengths:**

1. The proposed method is novel and solid theoretical. This work analyzes currently popular semi-linear samplers and different prediction targets and proposes a more general PC-based sampler. Contributions include:
    - unifies different prediction according to a dual-prediction parameterization.
    - uses a learned change of variables in semi-linear family for integration
    - second order residual adjustment for p1c2

2.  Experimental sufficient. This work provides strong, multi-backbone results on FID. And there's enough ablations over predictor/corrector order and parameter freedoms

3. Limitations are acknowledged.

Overall, this work proposed a well-motivated and clearly articulated solvers, especially for few-step sampling.

**Weaknesses:**

1. Personally, the article writing is somewhat difficult to follow up on. The overall structure is somewhat confusing to me.

2. The description of the parameter training part is insufficient. Especially the training process.

3. The discussion of details for certain algorithms lacks intuitive understanding.

**Questions:**

1. About the training process, my current understanding is that after sampling M-1 steps, we directly optimize the corresponding 10*(M-1) groups of parameters with a set of labels or captions? Once these parameters are trained, the parameters can be reused for this backbone? Can the author provide a more specific training process?

2. Why classification objectives beat regression is not deeply unpacked. The paper shows a V-shaped accuracy–FID trend and argues “reaching the right decision region” is enough, but it stops short of a deeper mechanism: when/why classification consistently wins across backbones/NFE, and under what conditions it might overfit decision boundaries.

3. Consistency vs. transfer across backbones/settings.
It observes that learned parameter curves look similar across NFEs for a given backbone, yet recommends training per backbone/setting. Readers may wonder about the practical transfer value of those similarities (warm-starts, NFE interpolation, shared parameterizations), which isn’t explored methodologically.

4. Fig2 is not mentioned in the article. And I feel that the Sec 4 and 5 are somewhat abrupt logically, it seems subsection of Sec2?


Overall, my main concern is 1. Please clarify the approach for obtaining parameters for all steps. If my concern addressed, I would be willing to raise my score.

---

> ### Author Response · Authors · 2025-11-21
> **Authors' response (1/2)**
>
> We thank the reviewer for the insightful comments and suggestions. We address all questions in the review below.
>
> **Can the author provide a more specific training process?**
>
> We recognized that the description of the training process for classification-based learning was insufficient, so we added *Algorithm 2* on *p.6* of the revised paper.
> To explain it more clearly, we refer to this method more specifically as hard-label classification, provide a schematic comparison with other learning methods in *Fig. 4*, and strengthen the explanation in *Sec. 5.2*.
>
> All Dual-Solver parameters $\phi$ are trained as follows: at each training iteration, we sample noise $\mathbf{x}_T$ and a class label $y$, run the sampling process to obtain a latent $\mathbf{x}_0$, pass it through the decoder and classifier to obtain class probabilities $p$, and then update $\phi$ by minimizing the cross-entropy between $p$ and the class label $y$.
> Training is performed for 20k steps using minibatches of size 10, where each minibatch consists of randomly sampled noises and class labels.
> After training, the learned parameters can be used consistently for the fixed backbone, CFG, and NFE setting they were trained for.
>
> **When/why classification consistently wins across backbones/NFE, and under what conditions it might overfit decision boundaries.**
>
> To better answer the question, “When and for which backbones is classification-based learning better?”, we additionally trained all backbones for NFEs from 3 to 9 and report the resulting FID and CLIP scores in *Fig. 5*.
> From these results, we observe that classification-based learning yields better FID and CLIP scores than regression-based solvers across all NFEs and backbones, except for GM-DiT at NFE = 7, 8, and 9.
>
> To provide a more detailed answer to the question, “Why is classification-based learning better than regression?”, we substantially revised *Sec. 5 Solver Parameter Learning*. We group existing regression-based approaches into trajectory / sample / feature regression, and introduce our proposed classification-based learning, which we further develop into soft-label and hard-label variants. Through this reorganization, we provide a detailed motivation for adopting classification-based learning.
>
> In diffusion models, prior work on timestep optimization [1] and distillation models [2] has observed that sample regression does not necessarily correlate with visual quality—typically measured by FID—and therefore uses feature-regression objectives such as LPIPS [3].
> We view these approaches as progressively increasing the abstract level of the matching objective space (from sample space to feature space).
> Extending this idea, we propose moving one step further to classification-based objectives, which compare samples at an even higher semantic level.
> Consistent with this perspective, *Table 3* shows a clear trend in the low-NFE regime (NFE = 3, 5): as we move from trajectory/sample regression to feature regression and then to our classification-based learning, FID decreases substantially.
> Since FID measures distance between distributions—rather than comparing samples directly (e.g., L1/L2/PSNR) or features directly (e.g., LPIPS)—we argue that classification objectives provide a more suitable surrogate than regression objectives.
>
> That said, classification-based learning may risk biasing the learned distribution toward a particular classifier.
> Indeed, *Fig. 6* shows that FID degrades when the classifier accuracy is either too high or too low.
> Therefore, we find that classifiers with modest accuracy align best with FID.
>
> **Readers may wonder about the practical transfer value of those similarities (warm-starts, NFE interpolation, shared parameterizations), which isn’t explored methodologically.**
>
> After observing that the parameter curves exhibit similar trends across different NFEs within the same backbone, we conducted an additional experiment that interpolates parameters learned at two different NFEs and applies them to other NFEs. The setup and results are reported in *Sec. 6.3 Parameter Interpolation across NFEs*. Although the interpolated parameters do not match the performance of parameters optimized separately for each NFE, they still yield competitive quality that consistently outperforms competing solvers.
>
> [1] Tong, Vinh, et al. "Learning to discretize denoising diffusion odes." arXiv preprint arXiv:2405.15506 (2024).
>
> [2] Song, Yang, et al. "Consistency models." (2023).
>
> [3] Zhang, Richard, et al. "The unreasonable effectiveness of deep features as a perceptual metric." Proceedings of the IEEE conference on computer vision and pattern recognition. 2018.

---

> > ### Author Response · Authors · 2025-11-21
> > **Authors' response (2/2)**
> >
> > **Fig2 is not mentioned in the article. And I feel that the Sec 4 and 5 are somewhat abrupt logically, it seems subsection of Sec2?**
> >
> > We revised *Fig. 2* to better match the paragraph on *discretization discrepancy* in *Sec. 2.2*, and added an explicit reference to *Fig. 2* in that paragraph.
> > *Sec. 4.1* and *Sec. 5.1* do contain some background material, so they could be moved to *Sec. 2*; however, doing so would make *Sec. 2* overly long and potentially fragmented, so we chose not to change their placement.
> > Instead, we added introductory text to *Sec. 4* and *Sec. 5* to smooth the narrative flow and avoid abrupt transitions.
> > Thank you for the careful reading and helpful suggestion.

---

> > ### Comment · Reviewer_qooq · 2025-11-27
> >
> > I thank the author for the response. Most of my concerns have been addressed.
> >
> > My current doubt is whether the class label in the training process can be replaced with other conditions? This would limit the generality of the method.
> >
> > But overall I am willing to raise my rating to 6, lean to acceptance.

---

> > > ### Author Response · Authors · 2025-11-28
> > >
> > > As you correctly pointed out, the hard-label classification method is restricted to conditional generation. However, in many practical scenarios such as text-to-image generation, it is standard to specify the desired target in advance, so we do not regard this as a major limitation in practice.
> > >
> > > For unconditional generation, one can instead use the soft-label classification method described in Sec. 5.2. As shown in Table 3, this approach yields clear FID gains over existing regression-based methods in the low-NFE regime (NFE = 3, 5). We also expect that, for NFE = 7 and 9, exploring appropriate classifier choices would further improve performance. A drawback is that, unlike hard-label classification, the soft-label variant requires a teacher sampler. We will further investigate soft-label classification and strengthen the corresponding discussion in the camera-ready version.
> > >
> > > Although this idea is still at a conceptual stage, we may consider learning solver parameters for unconditional generation by replacing the classifier with a pretrained autoregressive model, VAE, or flow-based network and maximizing log-likelihood or an ELBO (evidence lower bound) as a promising direction for future work.
> > >
> > > We sincerely thank you for the thorough review and for the constructive discussion and suggestions.

---

### Official Review · Reviewer_g7wy · 2025-10-29

**Soundness:** 3
**Presentation:** 3
**Contribution:** 2
**Rating:** 6
**Confidence:** 4

**Summary:**

This paper introduces Dual-Solver, which accelerates the sampling of diffusion models using learnable parameters $\gamma$, $\tau$, and $\kappa$. This framework incorporates and extends many existing multi-step and learning-based solvers, and abandons high-NFE (Number of Function Evaluations) teacher trajectories; instead, it updates the learnable parameters via classification loss or CLIP loss. The paper reports experimental results on ImageNet class-conditional and text-to-image diffusion models, effectively demonstrating the sampling performance of Dual-Solver under low NFE.

**Strengths:**

1. It constructs a generalized ODE solver based on a "predictor-corrector" structure. Through explicit interpolation of $\gamma$ and domain transformation via $\tau$, it unifies noise/data/velocity prediction methods within a single framework.
2. It proposes a solver parameter update method based on classification loss or CLIP loss, abandoning the parameter update learning strategy that relies on teacher trajectories. This reduces computational overhead and improves performance under low NFE.
3. The paper features rigorous mathematical derivations, covering the mathematical foundations of dual prediction and log-linear domain mapping. It also provides a complete proof of the second-order local truncation error.

**Weaknesses:**

1. The method is derived based on second-order accuracy, but it does not discuss potential issues when extending to higher-order schemes, which limits the further performance improvement of Dual-Solver.
2. It lacks comparative results with current advanced learning-based solvers, such as EPD-Solver[1] and AMED-Solver[2].
3. The parameter update of Dual-Solver relies on the cross-entropy loss of a classifier or CLIP loss, which means the parameter update of Dual-Solver will fail for unguided conditions.

[1] Zhu B, Wang R, Zhao T, et al. Distilling Parallel Gradients for Fast ODE Solvers of Diffusion Models[C]//Proceedings of the IEEE/CVF International Conference on Computer Vision. 2025: 19557-19566.

[2] Zhou Z, Chen D, Wang C, et al. Fast ode-based sampling for diffusion models in around 5 steps[C]//Proceedings of the IEEE/CVF Conference on Computer Vision and Pattern Recognition. 2024: 7777-7786.

**Questions:**

1. How does Dual-Solver perform compared to current advanced learning-based solvers (e.g., EPD-Solver[1], AMED-Solver[2])?
2. The paper proposes that using a "weak classifier" with lower accuracy can alleviate the problem of reduced sample diversity, but the results in Table 7 do not seem to clearly demonstrate the effectiveness of this strategy. Could you provide further explanation?
3. The paper learns parameters for each step but does not use ablation studies to quantify the effect of globally shared parameterization. Given that shared parameters can effectively reduce the number of learned parameters, could the authors conduct more detailed comparative experiments?

[1] Zhu B, Wang R, Zhao T, et al. Distilling Parallel Gradients for Fast ODE Solvers of Diffusion Models[C]//Proceedings of the IEEE/CVF International Conference on Computer Vision. 2025: 19557-19566.

[2] Zhou Z, Chen D, Wang C, et al. Fast ode-based sampling for diffusion models in around 5 steps[C]//Proceedings of the IEEE/CVF Conference on Computer Vision and Pattern Recognition. 2024: 7777-7786.

---

> ### Author Response · Authors · 2025-11-21
> **Authors' response**
>
> We thank the reviewer for a careful evaluation of our work. We respond to each question below.
>
> **How does Dual-Solver perform compared to current advanced learning-based solvers (e.g., EPD-Solver[1], AMED-Solver[2])?**
>
> We believe that the BNS-Solver[3] and DS-Solver[4], which we include as competing models in our paper, are sufficient as current advanced learning-based solvers.
> BNS-Solver is a follow-up to BST-Solver[5], which was presented as a spotlight paper at ICLR 2024, and BNS-Solver itself was presented at ICML 2024.
> DS-Solver was also presented at ICML 2025, reflecting the most recent advances in learning-based solvers.
> We also agree that EPD-Solver[1] and AMED-Solver[2] are meaningful comparison targets.
> However, AMED-Solver relies on CNN-style bottleneck features, which do not have a direct counterpart in our Transformer-based backbones (DiT, GM-DiT, SANA, and PixArt-α), so applying AMED-Solver to our setting is not straightforward.
> In addition, EPD-Solver is a parallelism-based method, making a fair comparison under the same setting nontrivial.
> So far, we have focused on additional ablation studies, and we will consider these solvers during the remaining rebuttal period and reflect them in the revised version as feasible.
>
> **The paper proposes that using a "weak classifier" with lower accuracy can alleviate the problem of reduced sample diversity, but the results in Table 7 do not seem to clearly demonstrate the effectiveness of this strategy. Could you provide further explanation?**
>
> In the revised paper, *Sec. 6.2 (D) Classifier model selection* analyzes the relationship between FID and classifier accuracy. FID is a metric that jointly captures the mean and covariance in feature space, but it does not explicitly disentangle diversity-related behavior.
> Therefore, in *Appendix D Classifier Accuracy vs. Sample Quality*, we provide a more detailed analysis by separating sample quality into precision and recall.
> Recall can be interpreted as reflecting diversity, and we observe a mild V-shaped trend at NFE = 3 and 9, with the best results achieved by classifiers ranked around 14–16.
>
> **The paper learns parameters for each step but does not use ablation studies to quantify the effect of globally shared parameterization. Given that shared parameters can effectively reduce the number of learned parameters, could the authors conduct more detailed comparative experiments?**
>
> We thank you for suggesting the idea of sharing parameters globally.
> We conducted additional ablation experiments on global parameter sharing and added the corresponding results to *Sec. 6.2 (B)* and *Table 2*.
> Overall, using global parameters still yields fairly low cross-entropy, but it does not match the performance of the fully learnable setting.
> We view this as a useful option when a smaller number of parameters is desired.
>
> [1] Zhu B, Wang R, Zhao T, et al. Distilling Parallel Gradients for Fast ODE Solvers of Diffusion Models[C]//Proceedings of the IEEE/CVF International Conference on Computer Vision. 2025: 19557-19566.
>
> [2] Zhou Z, Chen D, Wang C, et al. Fast ode-based sampling for diffusion models in around 5 steps[C]//Proceedings of the IEEE/CVF Conference on Computer Vision and Pattern Recognition. 2024: 7777-7786.
>
> [3] Shaul, Neta, et al. "Bespoke non-stationary solvers for fast sampling of diffusion and flow models." arXiv preprint arXiv:2403.01329 (2024).
>
> [4] Wang, Shuai, et al. "Differentiable Solver Search for Fast Diffusion Sampling." arXiv preprint arXiv:2505.21114 (2025).
>
> [5] Shaul, Neta, et al. "Bespoke solvers for generative flow models." arXiv preprint arXiv:2310.19075 (2023).

---

> > ### Comment · Reviewer_g7wy · 2025-11-28
> > **Official Comment by Reviewer g7wy**
> >
> > Thank you for the authors' efforts. Most of my concerns have been addressed, so I decide to keep the score unchanged and lean towards a weak accept.

---

### Official Review · Reviewer_TpbM · 2025-11-01

**Soundness:** 4
**Presentation:** 3
**Contribution:** 3
**Rating:** 6
**Confidence:** 4

**Summary:**

The paper proposes Dual-Solver, a learned predictor–corrector sampler for diffusion/flow-matching models that targets the low-NFE regime. It introduces three per-step learnable knobs: γ (interpolates prediction type across noise/data/velocity), τ (a log–linear domain transform that interpolates between λ- and ρ-domains), and κ (a residual term that preserves second-order local accuracy). The method keeps second-order local truncation error, learns the timestep schedule, and trains all solver parameters via a classification-based objective that uses a frozen classifier.

**Strengths:**

1. Unified, principled parameterization. γ cleanly bridges noise/data/velocity predictions; τ smoothly interpolates λ↔ρ domains; κ adjusts second-order residuals

2. Strong low-NFE results across backbones. Consistent FID/CLIP gains at ≤6 steps on DiT, SANA, PixArt-α, and GM-DiT.

3. Classification-based training removes the need for high-NFE teacher samples, reducing preparation cost and directly optimizing a perceptual proxy.

**Weaknesses:**

1. Training introduces ten learnable parameters per step (in addition to a learned schedule), which could increase brittleness or overfitting to specific backbones and guidance settings. Also, does the method require separate training runs for different NFE targets?

2. Because CE/CLIP is computed with a frozen classifier, the optimization might bias samples toward ‘classifier-friendly’ patterns. How do the authors mitigate proxy-objective leakage and ensure alignment with the intended generative quality?

**Questions:**

Does the method require separate training runs for different NFE targets?

---

> ### Author Response · Authors · 2025-11-21
> **Authors' response**
>
> We thank the reviewer for the thoughtful and constructive review. Below, we address the questions raised.
>
> **Does the method require separate training runs for different NFE targets?**
>
> We generally recommend training solver parameters separately for each backbone, guidance scale, and target NFE.
> However, we also find that parameters trained at other NFEs can be reused via simple linear interpolation: when we apply interpolated parameters to an unseen NFE, they still achieve reasonable performance and outperform baseline solvers.
> We have added an explanation of this procedure and the corresponding results to *Sec. 6.3 Parameter interpolation across NFEs*.
>
> **How do the authors mitigate proxy-objective leakage and ensure alignment with the intended generative quality?**
>
> We acknowledge that training with a fixed classifier can bias the generated samples toward what the classifier prefers.
> This concern is also common in feature-regression objectives such as LPIPS [1].
> We have added this issue to *Sec. 6.2 (C) parameter learning methods*.
>
> In this paper, we use FID and CLIP score as our measures of generative quality; accordingly, we conducted the following procedure to identify a classifier that is best aligned with FID:
>
> - We sort pretrained models from TorchVision [2] by their top-5 classification accuracy.
> - We uniformly select 20 models from the highest to the lowest accuracy.
> - For each of these 20 classifiers, we train Dual-Solver parameters with the GM-DiT backbone at NFE = 3 and 9.
> - For each setting, we generate 10k samples and measure FID.
>
> From these experiments, we choose MobileNet_V3_Large as the common classifier for parameter learning with both DiT and GM-DiT, since it consistently produced the best FID at both NFE = 3 and 9.
>
> For the text-to-image task, we applied the same procedure to OpenCLIP [3] models and selected an RN101 variant as the final choice.
> The corresponding details and results are provided in *Sec. 6.2 (D) parameter learning methods* and Appendix D *Classifier accuracy vs. sample quality*.
>
> [1] Zhang, Richard, et al. "The unreasonable effectiveness of deep features as a perceptual metric." Proceedings of the IEEE Conference on Computer Vision and Pattern Recognition. 2018.
>
> [2] https://docs.pytorch.org/vision/main/models.html
>
> [3] https://github.com/mlfoundations/open_clip

---

### Author Response · Authors · 2025-11-21
**General Response**

We sincerely thank the reviewers for their thoughtful feedback and constructive comments.
We have incorporated the feedback into the revised manuscript, with major changes summarized below for your convenience.

*For ease of review, we also note that the major newly added sentences are highlighted in blue in the revised manuscript.*

1. **Additional Experiments**
- **Fig. 5, Main quantitative results:** We updated Fig. 5 by adding new experiments for SANA and PixArt-$\alpha$ at NFE = 7, 8, and 9.
- **Sec. 6.2 (B), Parameter settings for $(\gamma,\tau,\kappa)$:** We added experiments to evaluate the feasibility of using global parameters and updated the section accordingly.
- **Sec. 6.2 (C), Parameter learning methods:** We conducted additional experiments to more clearly explain the transition from regression-based to classification-based learning.
- **Sec. 6.3, Parameter interpolation across NFEs:** We added experiments to test the robustness of parameters across NFEs.

2. **Revisions and Clarifications**
- **Fig. 2, Euler updates for noise, velocity, and data predictions:** We improved Fig. 2 and clarified the accompanying explanation.
- **Alg. 2, Hard-label classification for parameter learning:** We added Algorithm 2 to clarify the overall process of classification-based learning.
- **Sec. 5, Solver Parameter Learning:** We discussed regression- and classification-based learning in more depth.
- **Fig. 4, Solver parameter learning methods:** As Sec. 5 was expanded, we updated Fig. 4 to reflect and compare the detailed learning methods.

We hope that the extensive revisions and additional experiments adequately address the reviewers’ concerns.
We sincerely appreciate the reviewers’ efforts and hope that the revised manuscript meets the standards for acceptance.

---

### Meta-Review · Area_Chair_iRWd · 2025-12-30

**Summary:**

Reviewers generally liked the novel contributions towards a novel and unified solver for reducing NFE's, the theoretical rigor, and the strong experimental results.

Reviewer concerns were not particularly concentrated into specific issues (which is positive), and cover the following main points:
- Problems associated with incorporating classifiers in training (TpbM W2, qooq Q2 and follow-up comment, g7wy W3 Q2).
- Lack of clarity in description without easy intuition (qooq W1-3 Q1-2,4).
- Risk of overfitting with increased parameters (TpbM W1, g7wy Q3).
- Missing comparisons to learning-based solvers of EPD-Solver and AMED-Solver (g7wy W2 Q1).
- On the need for dual predictions (8gbp Q2 and follow-up comment).

**Reviewer Concerns:**

The authors provided relatively comprehensive responses to the reviewer concerns and questions, and appear to have extensive revised their paper. Based on these changes and some of the subsequent reviewer comments, the AC considers that the bulk of concerns have been resolved.

Besides the predicted final reviewer scores unanimously leaning to accept, the AC also considers the authors' submission to be an interesting novel contribution with good results, and recognizes that the original submission had been improved with substantial revisions. Hence the recommendation is to accept the paper.

**Reviewer Scores:**

The original reviewer scores are 6, 6, 6, 4. In the reviewer comments, reviewer qooq had indicated a willingness to increase their score from 4 to 6, while reviewer g7wy proposed to keep their score unchanged at 6 (but also mentioned leaning weak accept, which may perhaps be interpreted as above marginal accept).

---

### Decision · Program_Chairs · 2026-01-26

Accept (Poster)